# Evaluation of Promising Areas for Biogas Production by Indirect Assessment of Raw Materials Using Satellite Monitoring

Oleksiy Opryshko [1], Nikolay Kiktev [1,*], Sergey Shvorov [1], Fedir Hluhan [1,2], Roman Polishchuk [1], Maksym Murakhovskiy [3], Taras Hutsol [4,5], Szymon Glowacki [6,7,*], Tomasz Nurek [8] and Mariusz Sojak [6]

1 Department of Automation and Robotic Systems, National University of Life and Environmental Sciences of Ukraine, Heroiv Oborony Str. 15, 03041 Kyiv, Ukraine; ozon.kiev@gmail.com (O.O.); sosdoc@i.ua (S.S.); arr55005@gmail.com (F.H.); polishchuk.r23@gmail.com (R.P.)
2 National Space Facilities Control and Test Center, Knyaziv Ostroz'kykh Str. 8, 01010 Kyiv, Ukraine
3 Department of Corporate Finance and Controlling, Kyiv National Economic University Named After Vadym Hetman, 54/A Beresteysky Avenue, 03057 Kyiv, Ukraine; maxzerno@ukr.net
4 Department of Machine Operation, Ergonomics and Production Processes, Faculty of Production and Power Engineering, University of Agriculture in Krakow, Balicka 116B, 30-149 Krakow, Poland; wte.inter@gmail.com
5 Department of Agricultural Engineering, Odesa State Agrarian University, Panteleimonivska Str., 65012 Odesa, Ukraine
6 Department of Fundamentals of Engineering and Power Engineering, Institute of Mechanical Engineering, Warsaw University of Life Sciences (SGGW), 02-787 Warsaw, Poland; mariusz_sojak@sggw.edu.pl
7 Ukrainian University in Europe–Foundation, Balicka 116, 30-149 Krakow, Poland
8 Department of Biosystem Engineering, Institute of Mechanical Engineering, Warsaw University of Life Sciences (SGGW), 02-787 Warsaw, Poland; tomasz_nurek@sggw.edu.pl
* Correspondence: nkiktev@ukr.net (N.K.); szymon_glowacki@sggw.edu.pl (S.G.)

**Abstract:** An important issue in the sustainable development of agricultural engineering today is the use of biogas plants for the production of electricity and heat from the organic waste of agricultural products and other low-quality products, which also contributes to the improvement of environmental safety. Traditional methods for assessing the apparent severity of the Roslynnytsia campaign based on statistics from the dominions proved to be ineffective. A hypothesis was proposed regarding the possibility of estimating the apparent biomass by averaging the indicators of depletion and assessing the $CH_4$ and CO emissions based on satellite monitoring data. The aim of this work is to create a methodology for preparing a raw material base in united territorial communities to provide them with electrical and thermal energy using biogas plants. The achievement of this goal was based on solving the following tasks: monitoring biomethane emissions in the atmosphere as a result of rotting organic waste, and monitoring carbon monoxide emissions as a result of burning agricultural waste. Experimental studies were conducted using earth satellites on sites with geometric centers in the village of Gaishin in the Pereyaslav united territorial community, the city of Ovruch in the Zhytomyr region, the Oleshkovsky Sands National Park in the Kherson region (Ukraine), and the city of Jüterbog, which is located in the state of Brandenburg and is part of the Teltow-Fläming district (Germany). The most significant results of this research involve the methodology for the preparation of the raw material base in the united territorial communities for the production of biogas, based on indirect measurements of methane and carbon dioxide emissions using the process of remote sensing. Based on the use of the proposed scientific and methodological apparatus, it was found that the location of the territory with the center in the village of Gaishin has better prospects for collecting plant raw materials for biogas production than the location of the territorial district with the center in the city of Ovruch, the emissions in which are significantly lower. From March 2020–August 2023, a higher CO concentration was recorded on average by 0.0009 $mol/m^2$, which is explained precisely by crop growing practices. In addition, as a result of the conducted studies, for the considered emissions of

methane and carbon monoxide for monitoring promising raw materials, carbon monoxide has the best prospects, since methane emissions can also be caused by anthropogenic factors. Thus, in the desert (Oleshkivskie Pisky), large methane emissions were recorded throughout the year which could not be explained by crop growing practices or the livestock industry.

**Keywords:** biomass; agricultural waste; remote monitoring; microclimate of the territory; satellite sensing; energy production; methane; carbon monoxide; algorithm; web application

## 1. Introduction

In the current trends of the global world, new threats arise, including towards energy and agriculture. This hinders the sustainable development of these industries. Such threats include individual and group terrorist activities and military conflicts between states, which significantly change the assessment of the physical security of the existing energy infrastructure. It should be noted that large power plants, especially nuclear ones, have some protection, while distribution points, substations, and power lines in areas of increased risk are more vulnerable to such threats. Awareness of these threats is investigated and described in the article by S. Kutjuns et al. [1].

The European Commission has introduced requirements stating that energy companies must implement urgent measures to counter these threats. Thus, the overall security of energy supplies can be strengthened using measures to counter cyberattacks, as shown in the article by Z. Wang and G. Chen [2]. Such situations were also modeled based on the results of the study by F. Yanine et al. [3], which predicted the emergence of microgrids with inherent problems of asymmetry in their autonomous generation and consumption. In conditions where a possible shortage of electricity occurs in an emergency situation, energy companies will be forced to implement a consumer hierarchy, as shown in S.D. Manshadi and M. Khodayar [4], as well as industrial energy storage systems, which are proposed in N. Padmanabhan et al. [5]. Such measures are effective for the emergency provision of critical facilities, where the cost of electricity is relatively unimportant. For the vast majority of remote facilities with lower requirements for the quality and continuity of energy consumption, it is advisable to implement the concept of energy-independent communities, in which, along with traditional generation, their own renewable sources are actively used.

The world has already accumulated extensive experience in implementing the concept of energy-independent communities, and along with the initial purely technical issues, organizational issues are also being addressed, namely economic K. Tian et al. [6] and legal D. D'Achiardi et al. [7], regarding the interaction between traditional and renewable generation. Since the introduction of renewable generation is a necessary condition for energy security in communities, research has been conducted on ways to compensate for uneven generation, which is inherent in photovoltaic and wind generation. The work of M.U. Khan et al. [8] in India and A. Zeinalzadeh et al. [9] in Australia show that the most appropriate way to quickly compensate for possible failures of such generation is to use gas generation.

For agricultural enterprises and in rural areas and settlements, biogas plants are a convenient and territorially close resource for obtaining biogas, as well as electrical and thermal energy. Different agricultural enterprises and territories use different methods to provide biogas plants with raw materials. These can be livestock waste as well as plant residues (straw), as described in the work of H. Ezz et al. [10]. Another option for obtaining

raw materials is the use of traditional and energy crops (sorghum silage), as shown in the study by P. Pochwatka et al. [11].

An additional reason for the implementation of biogas plants to ensure energy independence in agricultural communities is, according to K. Saravanan et al. [12] and V. M. Lapushkin et al. [13], the production of high-quality organic fertilizers as a by-product of biogas production.

Thus, for the sustainable development of the agricultural industry, it is advisable and effective to introduce biogas plants and include them in the complex of energy systems within territories. However, the issue of determining, searching for, and selecting raw material, as well as its availability, type, and location, is not sufficiently studied and researched. Even in the favorable climate of Thailand, according to S. Khotmanee and U. Pinsopon [14], for the cultivation of the energy crop Napier grass, logistically suitable marginal lands were selected from all possible options using satellite imagery.

Currently, the problem of energy independence is relevant for Ukraine and Poland; therefore, the development of methodological recommendations for the creation of a raw material base for biogas plants in settlements, territories, and agricultural enterprises was the goal of this study (Figure 1).

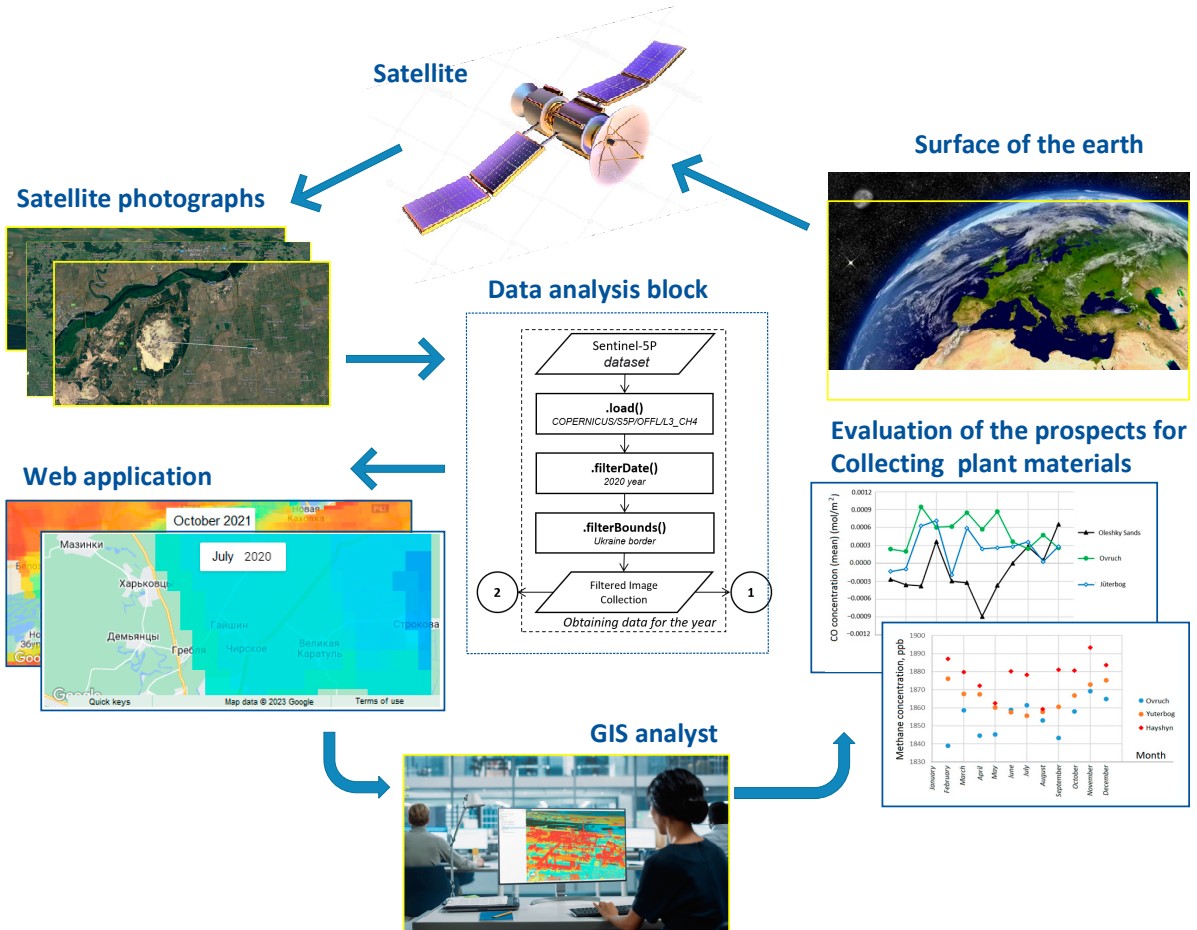

**Figure 1.** Structural diagram of the study.

For biogas synthesis, bird droppings, farmanimal manure, and plant biomass are used as raw materials. However, there is less information about the plant biomass necessary for organizing biogas production in each locality or district for both organizational and purely methodological reasons.

Some agricultural enterprises have their own biogas plants and they have tried to optimally use their resources to obtain maximum profit. The studies of V. Lysenko et al. [15] and D. Komarchuk et al. [16] are devoted to this issue. At the first stages of vegetation the condition of the plants was assessed, as a result of which areas were identified where plants were affected due to insufficient nitrogen nutrition, and therefore the value of the crop was reduced. Such plants can be used for biogas production, including in the spring period, when there is a shortage of raw materials.

However, the use of straw is not necessarily a developed practice. According to D.P. Biswas [17] and A. Anand et al. [18], rice straw and sugarcane are well suited for creating fuel brackets, while on the contrary, J. Li et al. [19] showed that corn straw is, in many cases, simply burned by farmers directly in the fields. According to X. Zhang et al. [20], part of the straw is deliberately smeared into the soil by farmers using modern agricultural technologies to enrich it with carbon. Smearing is not a mandatory practice in Ukraine, since it is then necessary to apply increased amounts of nitrogen fertilizers, the cost of which has increased in recent years. In case of incompleteness and limitations of the specified information, it may be advisable to use indirect methods of assessing prospective biomass.

An interesting technique is the use of remote sensing (RS) and geographic information systems (GIS) in solving the problem of solid waste management in connection with the ever-growing world population. The article by Indian researchers Sakhi et al. [21] describes the implementation of integrated solid waste management (ISWM) in connection with limited space for landfills. The use of RS methods allowed accurate mapping of waste generation and disposal sites, and GIS allowed for optimization of waste collection routes, selection of burial sites, and waste-to-energy projects.

For crop practices using satellite imaging, the sown crop and crop area can be established as shown in the work of A. Kayad et al. [22]. A new non-standard approach to the search for raw materials is shown in the study by S. A. Shvorov et al. [23]. The authors estimated the availability of biomass based on methane emissions from satellite imagery. Methane emissions are primarily due to the decay of organic matter; however, organic matter can also be burned, resulting in carbon monoxide (CO) emissions—accordingly, it is advisable to estimate the emissions of this reagent, which, according to Genevieve Plant et al. [24], can also be carried out using satellite imaging.

An analysis of the studies on this topic has allowed us to conclude that, when managing territories and agricultural enterprises, a valuable source of information on the available (unused) plant resources is via the remote assessment of the products formed during the destruction of organic matter. The result of its decomposition is the formation of methane, and the result of combustion is carbon monoxide, the remote monitoring of which is best assessed using satellite means.

## 2. Materials and Methods

### 2.1. Selection of Sites for Research

Further research will be devoted to the processing and analysis of images from space satellites. For this purpose, the following territories have been identified:

(A) Gaishin. The pilot project on energy independence identified the Pereyaslavsky district (27,000 inhabitants in 2019), the geographical center of which is the village of Gaishin. The district has significant areas of agricultural land, and, importantly, a developed agricultural infrastructure, as well as access to forest resources (Figure 2). A combination of factors, such as the location of the district and the resource base, are favorable for generation using biogas plants and organizing the district's activities in conditions of energy independence.

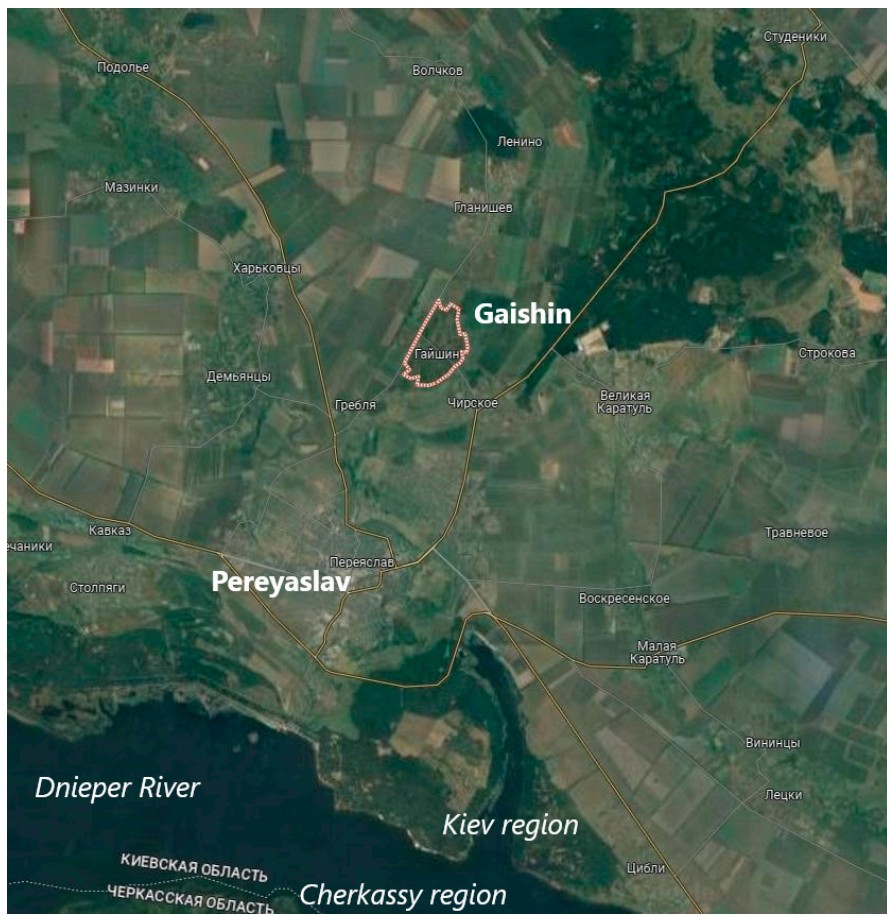

**Figure 2.** Satellite image of the Pereyaslavsky district of the city council (red dotted line—border of the village of Gaishin) (source—Google maps).

(B) Ovruch. The previous district (Pereyaslavsky) which we studied is located near the Dnieper River. In this location, methane and CO emissions may be affected by the natural destruction of organic matter formed directly in the river: algae and reeds. In the article by Y. Zhang et al. [25], it is noted that the volume of river biomass can be significant and even unstable due to its transport by water along the stream. To determine the amount of biomass created in the wild, we chose the city of Ovruch, Zhytomyr region (16,000 inhabitants), which has a more continental location; there are no large water bodies nearby (Figure 3). This city has significant agricultural potential; located among forests, here, as in Gaishin, the impact of emissions caused by woody organic matter is possible.

The Ukrainian government has passed laws that prohibit the burning of crop residues in fields. The laws are aimed at improving the ecological state and preserving soils, because valuable soil-forming microorganisms die as a result of fires. However, in Ukraine the farm owners and workers often break the law and burn organic residues because this makes their work in the fields easier. Therefore, we proposed to explore the next location in Germany, where environmental legislation is traditionally more strictly enforced.

(C) Jüterbog. The city is located 72 km from the center of Berlin (60 km from the outskirts) in the state of Brandenburg; it is part of the Teltow-Fleming district. The population of Jüterbog is 12,668 people. The territorial community has large areas of arable land, which are surrounded by forests with significant organic potential (Figure 4). The distance to the nearest metropolis (Berlin) is approximately the same as from Gaishin to Kyiv. There are no large rivers or lakes near Jüterbog.

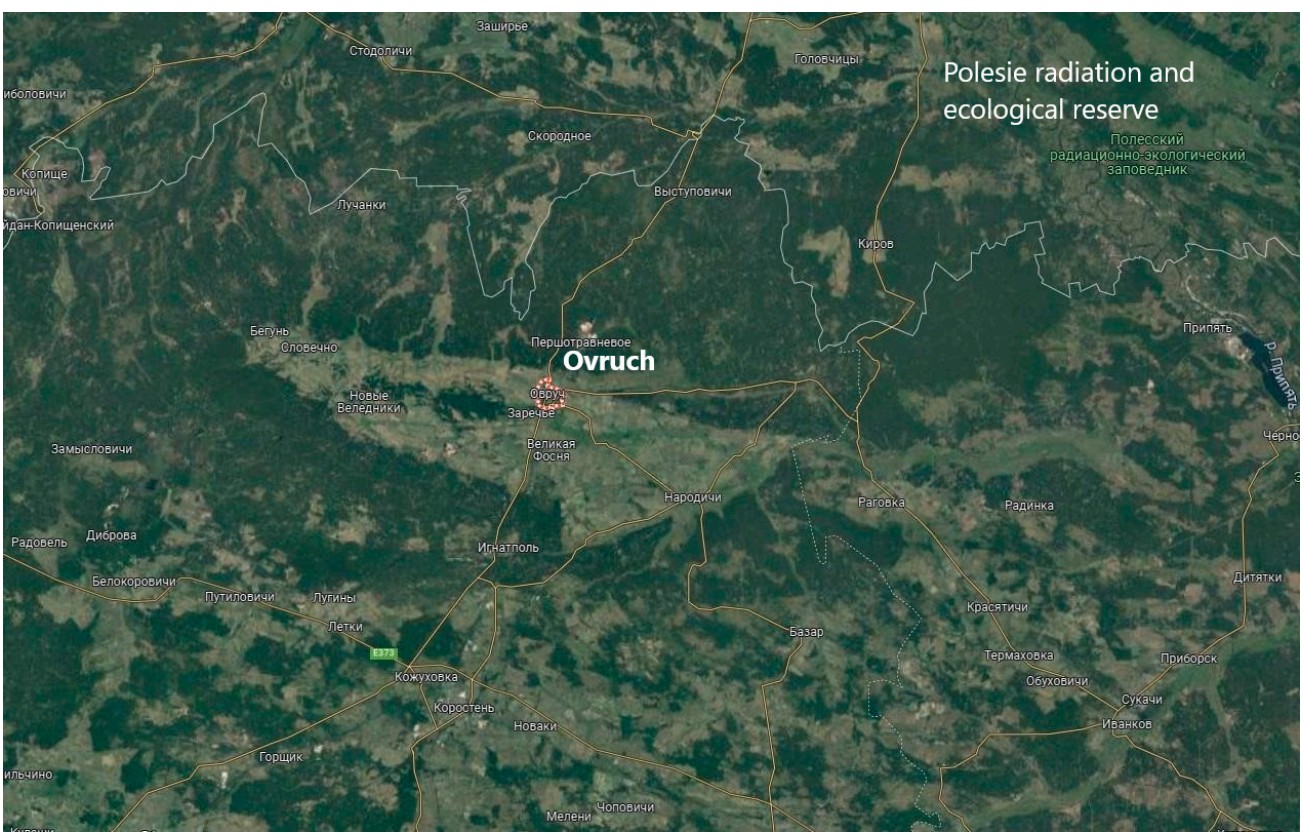

**Figure 3.** Satellite image of the Ovruch district (red dotted line—border of Ovruch city) (source—Google maps).

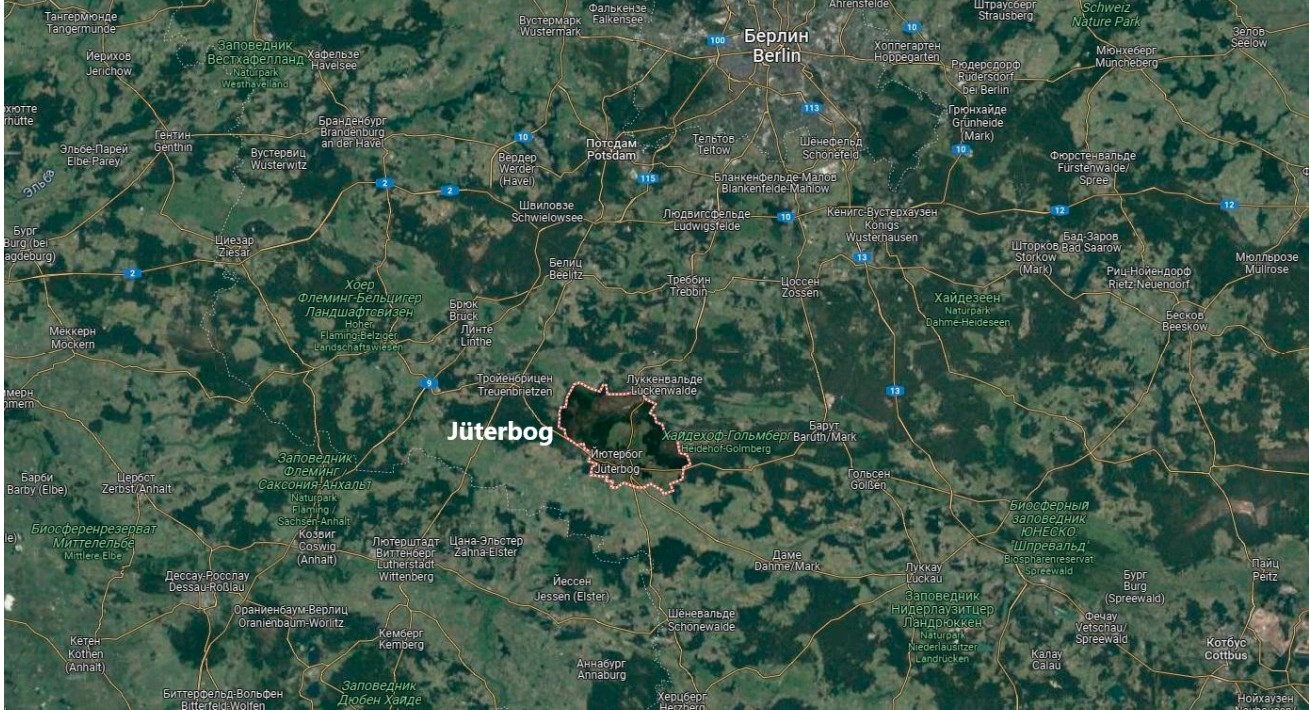

**Figure 4.** Satellite image of the surroundings of the city of Jüterbog (city border is highlighted with a red dotted line) (source—Google maps).

(D) Oleshky Sands National Nature Park. The following location is proposed by us as a control point, because it represents a desert with practically no vegetation and probably no CH₄ and CO emissions. This park is located in the Kherson region of Ukraine, 40 km

from the regional center of Kherson. The park is built on two Nizhny Dnieper arenas: Kozachelagerska and Chalbaska. Its total area is 11.7 thousand hectares; the land is state-owned, of which 2.8 thousand hectares come without the right to withdraw from land users (Figure 5).

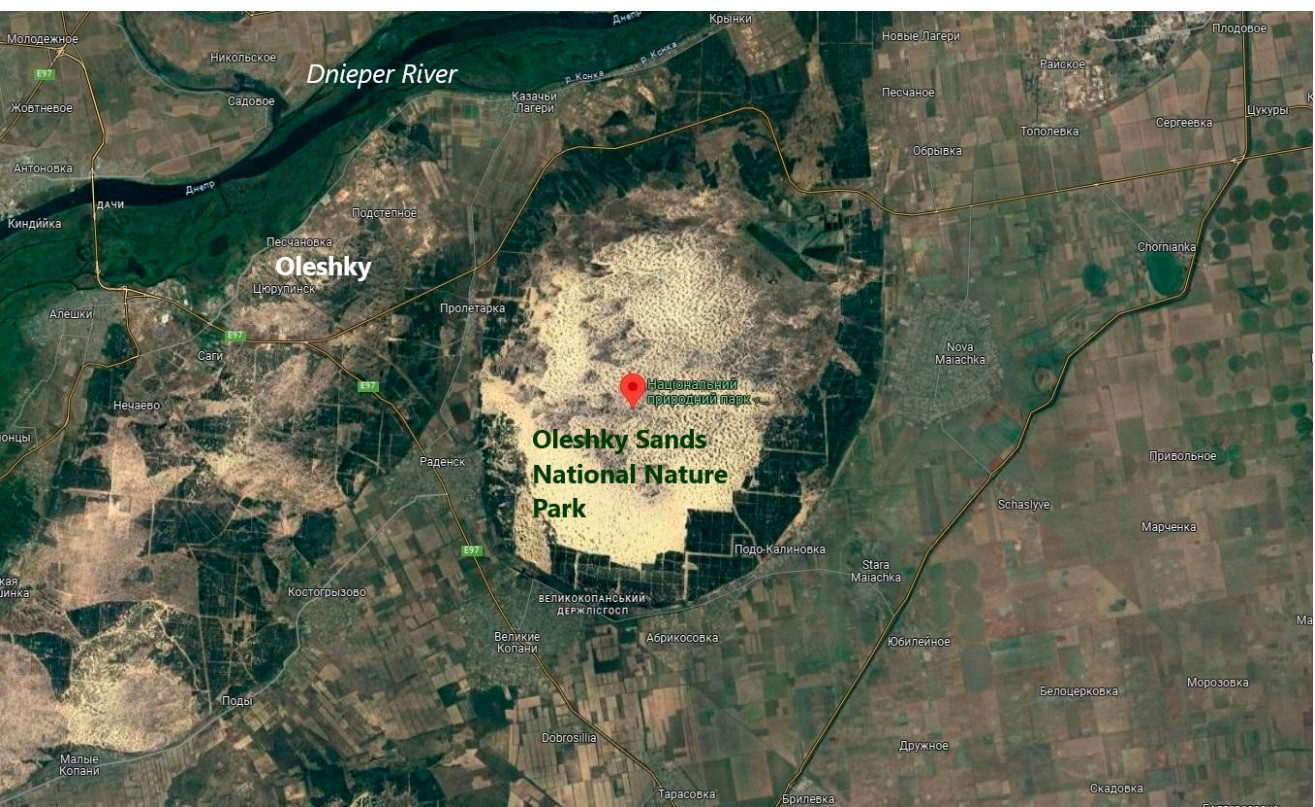

**Figure 5.** Satellite image of Oleshky Sands National Nature Park (image obtained from the Google maps internet service).

### 2.2. Methodology of the Experiment

When conducting this research, the authors used programs developed by specialists from the National Center for Space Control and Testing of Ukraine (https://spacecenter.gov.ua/, accessed on 25 February 2025). The software used is described in more detail in the previous study by S.A. Shvorov et al. [23] on the use of satellite data for monitoring urbanized technologies. The issue of the repeatability of positioning in different years was controlled in the program by fixing the coordinates of the points that were displayed by the web application (Figure 6).

The experiment consisted of predicting spatiotemporal changes in the atmospheric CO and $CH_4$ content in 2020–2022. The Sentinel 5P air monitoring data from the EU Copernicus program (https://www.copernicus.eu/en/about-copernicus, accessed on 25 February 2025) were used for this purpose. The goal of the Sentinel 5P mission, launched in 2017, is to perform high-space–time measurements to analyze the chemical composition of the Earth's atmosphere, as well as to monitor and forecast climate change.

The methodology of using satellites to measure CO and $CH_4$ is described in [23]. Methane and carbon monoxide are measured using the TROPOspheric Monitoring Instrument (TROPOMI) spectrometer installed on the Sentinel-5P satellite. According to the ESA (European Space Agency) documentation, the average deviation of carbon monoxide and methane is approximately 10% and 1.5%, respectively, which indicates high data reliability. The measurement methods are described in [26]. The data for analysis are converted to level 3 (L3) using the HARP (High Altitude Research Project) tools described in [27], installed

on a cloud platform. An example of working with polyurethane encoding is given in Appendix A. The Sentinel-5P dataset was used to obtain high-resolution methane images, filtering out data from 2020.

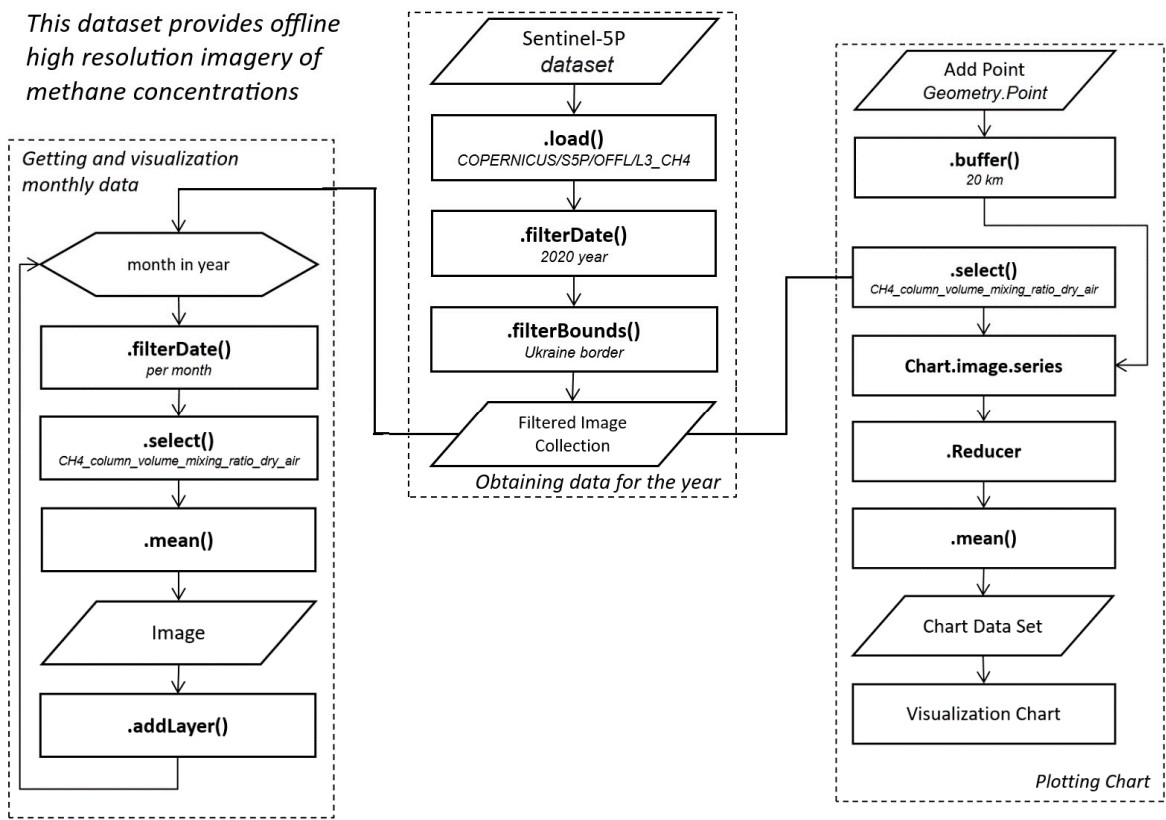

**Figure 6.** Block diagram of the algorithm for calculating spatiotemporal changes in the content of CO and $CH_4$ in the atmosphere.

### 2.3. Data Processing

This research was conducted using the techniques developed by specialists from the National Center for Control and Testing of Space Facilities of Ukraine (https://spacecenter.gov.ua/, accessed on 25 February 2025). Monitoring of methane and CO concentrations was carried out based on satellite monitoring data, which is obtained as part of the first mission dedicated to air monitoring; Sentinel-5P under the European Union (EU) Earth observation program—Copernicus (https://www.copernicus.eu/en/about-copernicus, accessed on 25 February 2025). The software is described in a previous study by S.A. Shvorov et al. [23] on urbanized technologies. We used data from the period January 2020–June 2023. Since satellite monitoring requires appropriate weather conditions, the number of measurements per month varies throughout the year; this research used the arithmetic average data per month. Since the data resolution is several $km^2$/pixel, positioning was carried out in the center of the study area. The repeatability of positioning in different years was monitored by fixing the coordinates of the points displayed in the web application (Figure 7).

The coordinates of the measurement points (longitude and latitude) are recorded in the left part of the interface. A certain number of measurements are performed every day, and the user can access information about these in a tabular format in the developed web application.

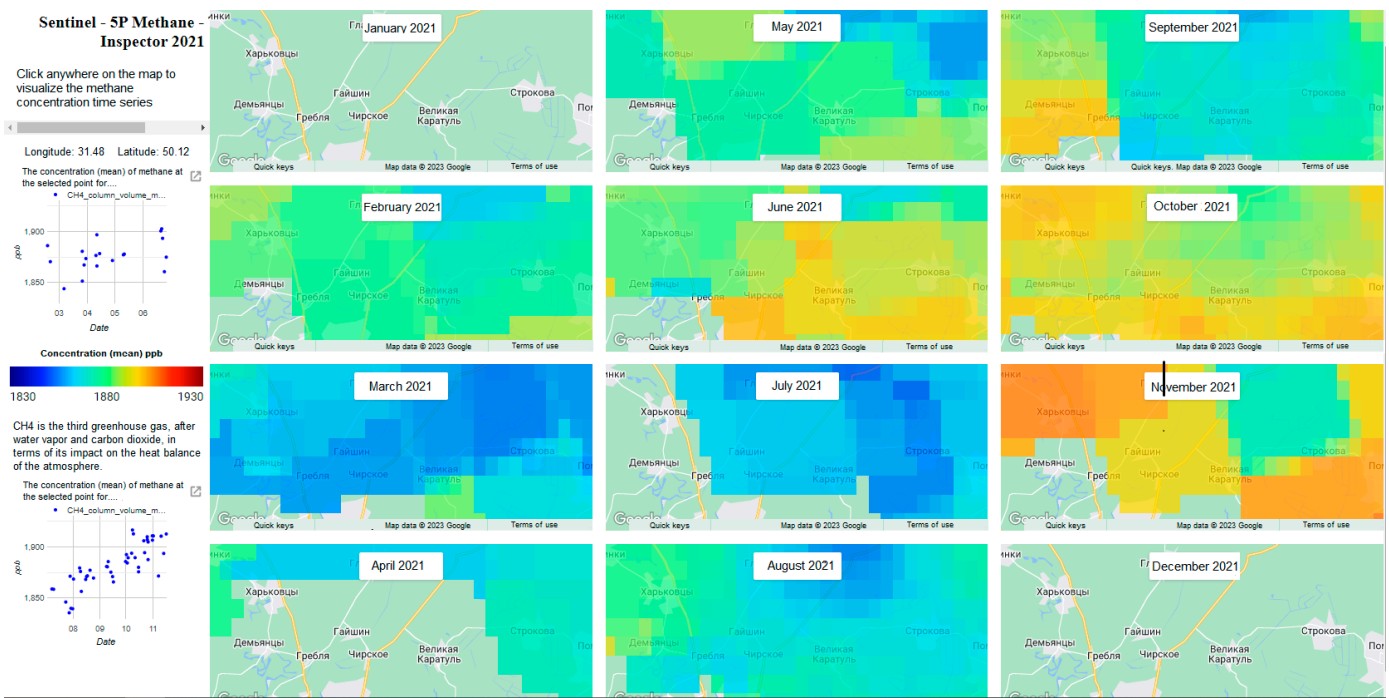

**Figure 7.** Web application interface for assessing spatiotemporal changes in the content of CH$_4$ in the atmosphere for the settlement of Gaishin in 2021.

## 3. Results and Discussion

### 3.1. Methane Emissions Assessment

Analysis of satellite images of the Oleshky Sands National Nature Park showed that methane emissions can still be observed over the area with no vegetation (sand) and no decay (Figure 8). The results are confirmed by observations over several years.

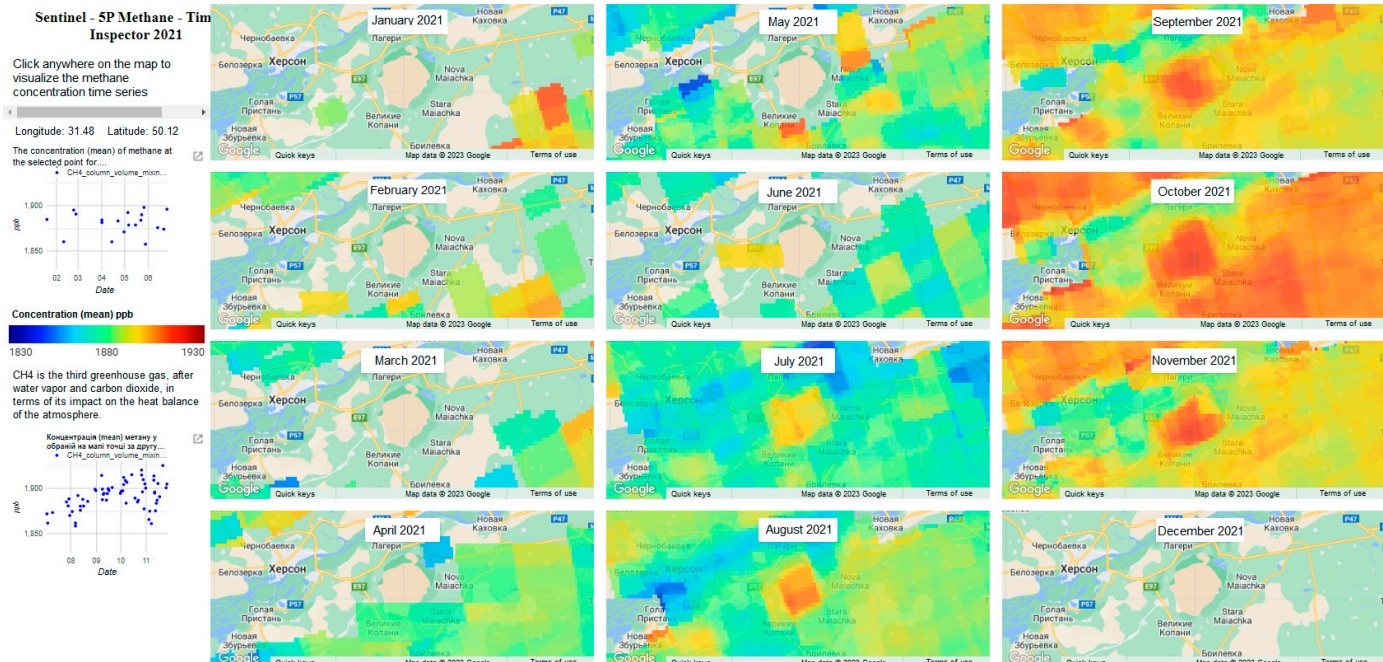

**Figure 8.** Methane emissions during 2021 in the Kherson region over the Oleshky Sands research site.

The "Novokamenka" site was additionally selected for research (due to the military conflict that began in this region in 2022). The new site is similar to the main one in its

natural characteristics, located 33 km from the Oleshky Sands location. The new location is intended for irrigated agriculture (Figure 9).

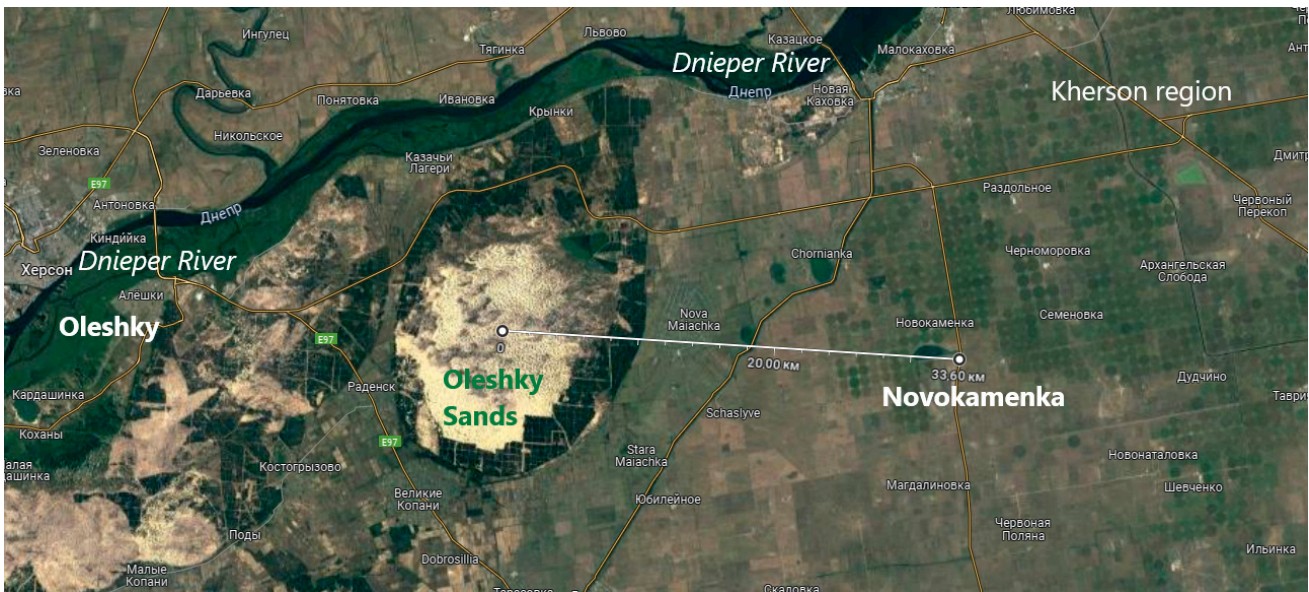

**Figure 9.** Satellite image of the location of the main and additional locations in the Kherson region, Oleshky Sands and Novokamenka, respectively.

The results of the satellite remote sensing showed that no methane emission sources were visually observed at the other study sites (Figure 10). The results of remote sensing are presented in Tables 1 and 2.

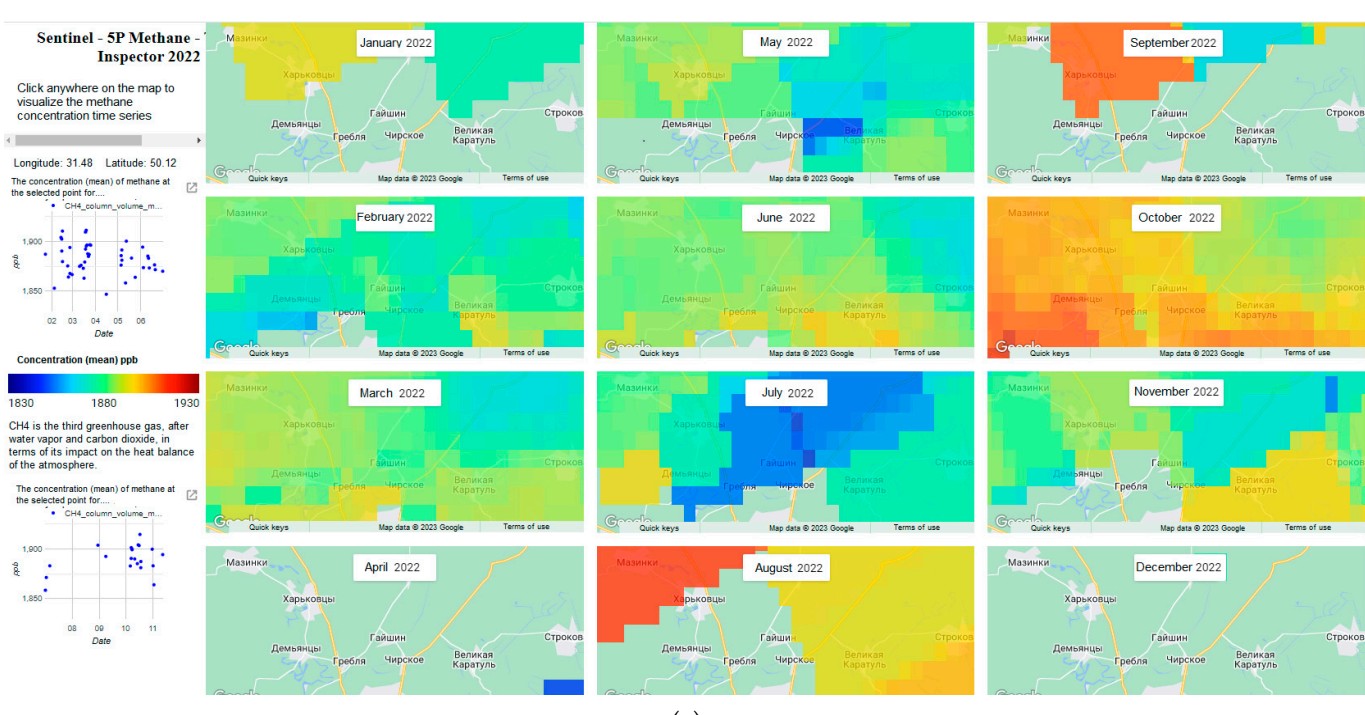

(**a**)

**Figure 10.** *Cont*.

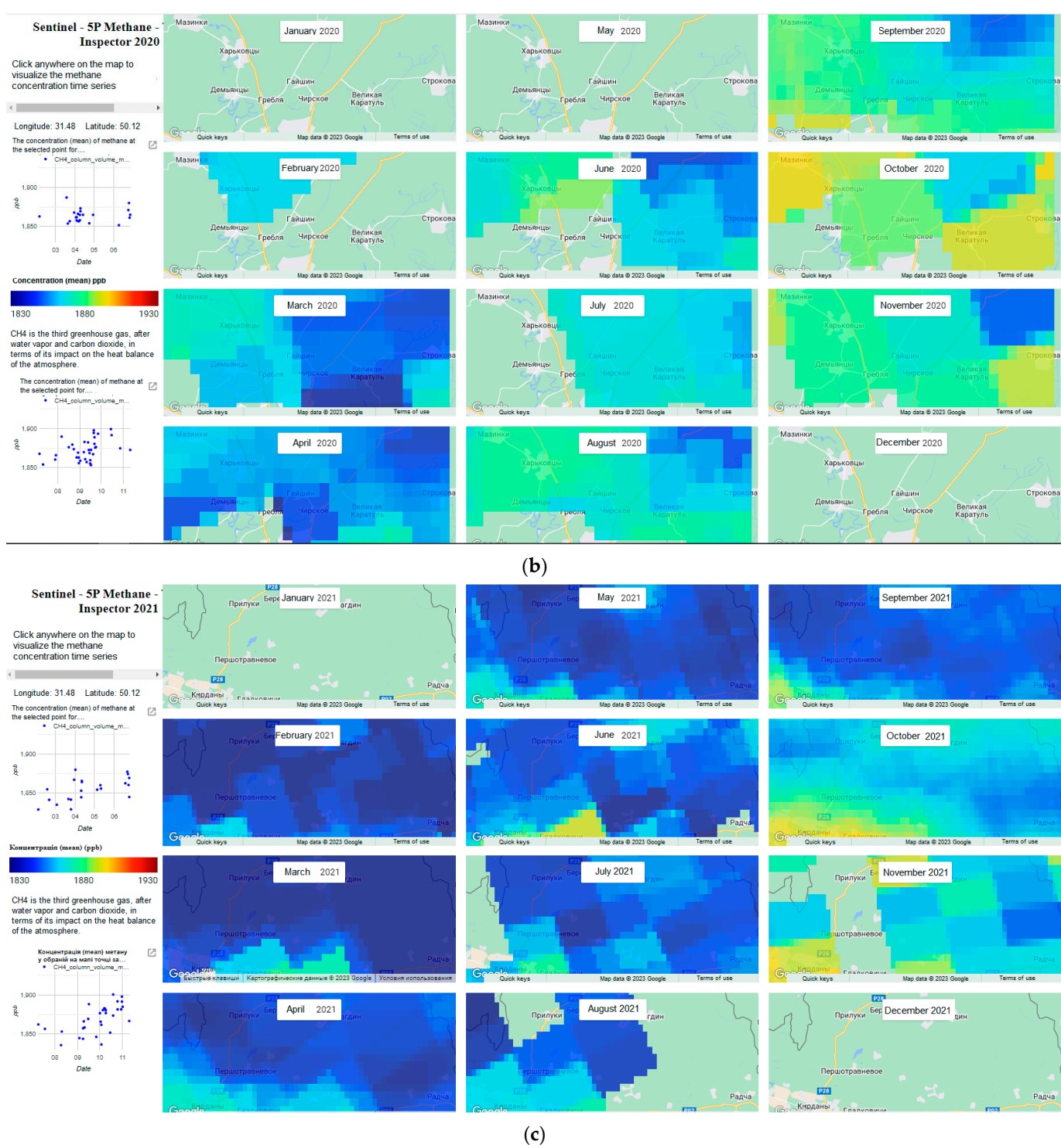

**Figure 10.** *Cont.*

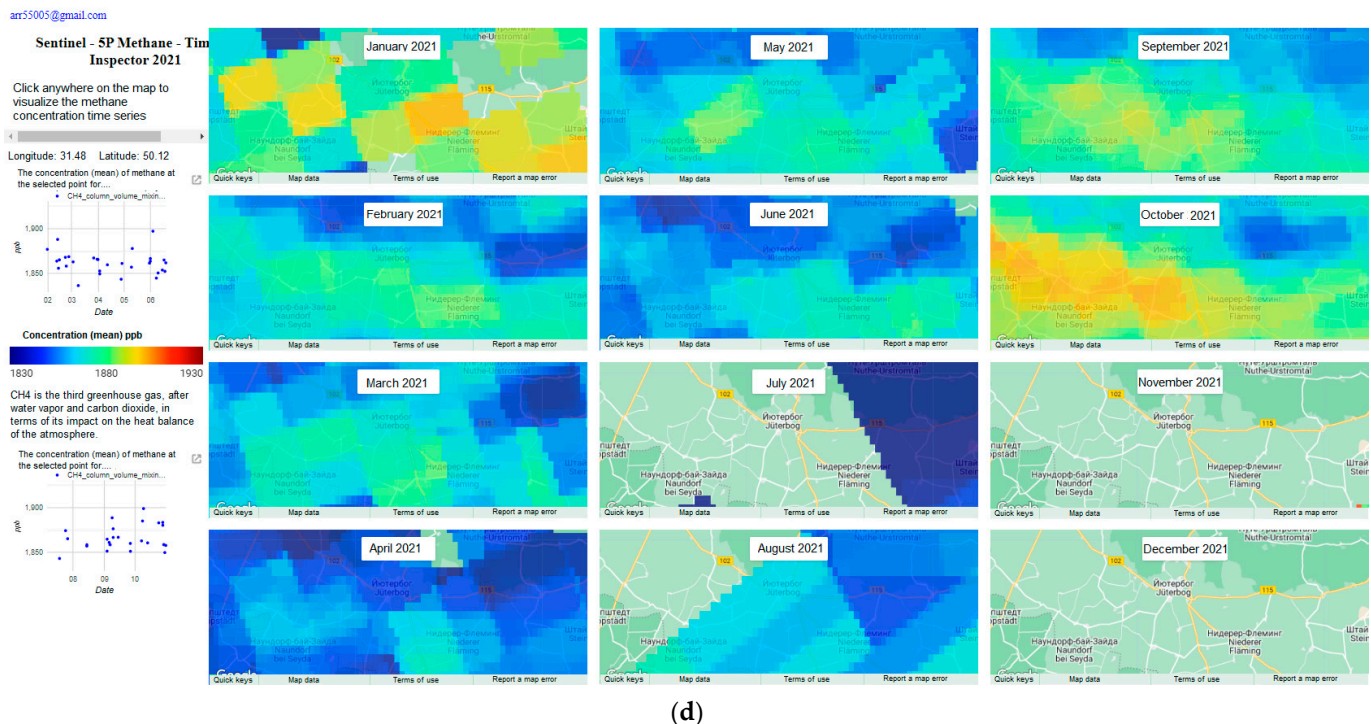

**Figure 10.** Remote sensing results—methane emissions at the research sites Gaishin 2022 and 2020 (**a**,**b**), Ovruch 2021 (**c**), and Jüterbog 2021 (**d**).

**Table 1.** Recorded methane emissions for the locations of Gaishin, Ovruch, and Jüterbog by year, ppb.

| | Gaishin | | | | Ovruch | | | | Jüterbog | | | |
|---|---|---|---|---|---|---|---|---|---|---|---|---|
| | **2020** | **2021** | **2022** | **2023** | **2020** | **2021** | **2022** | **2023** | **2020** | **2021** | **2022** | **2023** |
| 1 | | | 1887 | | 1839 | | | | | 1876 | | |
| 2 | 1862 | 1878 | 1882 | 1897 | 1845 | 1841 | 1865 | 1883 | 1837 | 1866 | 1877 | 1891 |
| 3 | 1866 | 1863 | 1888 | 1872 | 1823 | 1830 | 1868 | 1857 | 1852 | 1859 | 1878 | 1881 |
| 4 | 1859 | 1877 | 1846 | 1867 | 1821 | 1861 | 1830 | 1869 | 1849 | 1853 | 1859 | 1881 |
| 5 | | 1877 | 1880 | 1884 | | 1856 | 1846 | 1874 | 1831 | 1865 | 1860 | 1874 |
| 6 | 1865 | 1886 | 1878 | 1884 | 1845 | 1865 | 1870 | 1867 | 1829 | 1861 | 1859 | 1875 |
| 7 | 1861 | 1846 | 1870 | | 1836 | 1859 | 1864 | | 1874 | 1839 | 1860 | |
| 8 | 1870 | 1870 | 1903 | | 1849 | 1838 | | | 1864 | 1858 | | |
| 9 | 1873 | 1877 | 1892 | | 1843 | 1859 | 1872 | | 1870 | 1864 | | |
| 10 | 1888 | 1897 | 1895 | | 1853 | 1873 | 1882 | | | 1872 | 1874 | |
| 11 | 1872 | 1899 | 1880 | | 1828 | 1875 | 1892 | | 1863 | | 1888 | |

**Table 2.** Recorded methane emissions for the Oleshky Sands and Novokamenka locations by year, ppb.

| | Oleshky Sands | | | | Novokamenka | | | |
|---|---|---|---|---|---|---|---|---|
| | **2020** | **2021** | **2022** | **2023** | **2020** | **2021** | **2022** | **2023** |
| 1 | 1869 | 1884 | 1880 | | 1869 | 1895 | 1893 | 1891 |
| 2 | 1857 | 1882 | 1883 | 1900 | 1866 | 1889 | 1893 | 1898 |
| 3 | 1859 | | 1879 | 1880 | 1863 | 1882 | 1886 | 1885 |
| 4 | 1868 | 1876 | 1893 | | 1866 | 1884 | 1889 | |
| 5 | 1867 | 1882 | 1881 | 1889 | 1872 | 1880 | 1883 | 1888 |
| 6 | 1861 | 1882 | 1886 | 1898 | 1863 | 1878 | 1887 | 1891 |
| 7 | 1872 | 1875 | 1886 | | 1876 | 1878 | 1890 | |
| 8 | 1882 | 1881 | 1883 | | 1890 | 1886 | 1883 | |
| 9 | 1880 | 1896 | 1894 | | 1888 | 1895 | 1898 | |
| 10 | 1889 | 1900 | 1902 | | 1891 | 1903 | 1904 | |
| 11 | 1877 | 1892 | 1891 | | 1884 | 1895 | 1893 | |
| 12 | | | 1891 | | | | 1889 | |

### 3.2. Carbon Monoxide Emission Assessment

Remote sensing analysis has established that the recorded concentration of CO in the atmosphere is distributed more evenly (Figure 11), compared to methane. In this regard, we did not introduce additional locations for research. Remote sensing results for location by year are given in Tables 3 and 4.

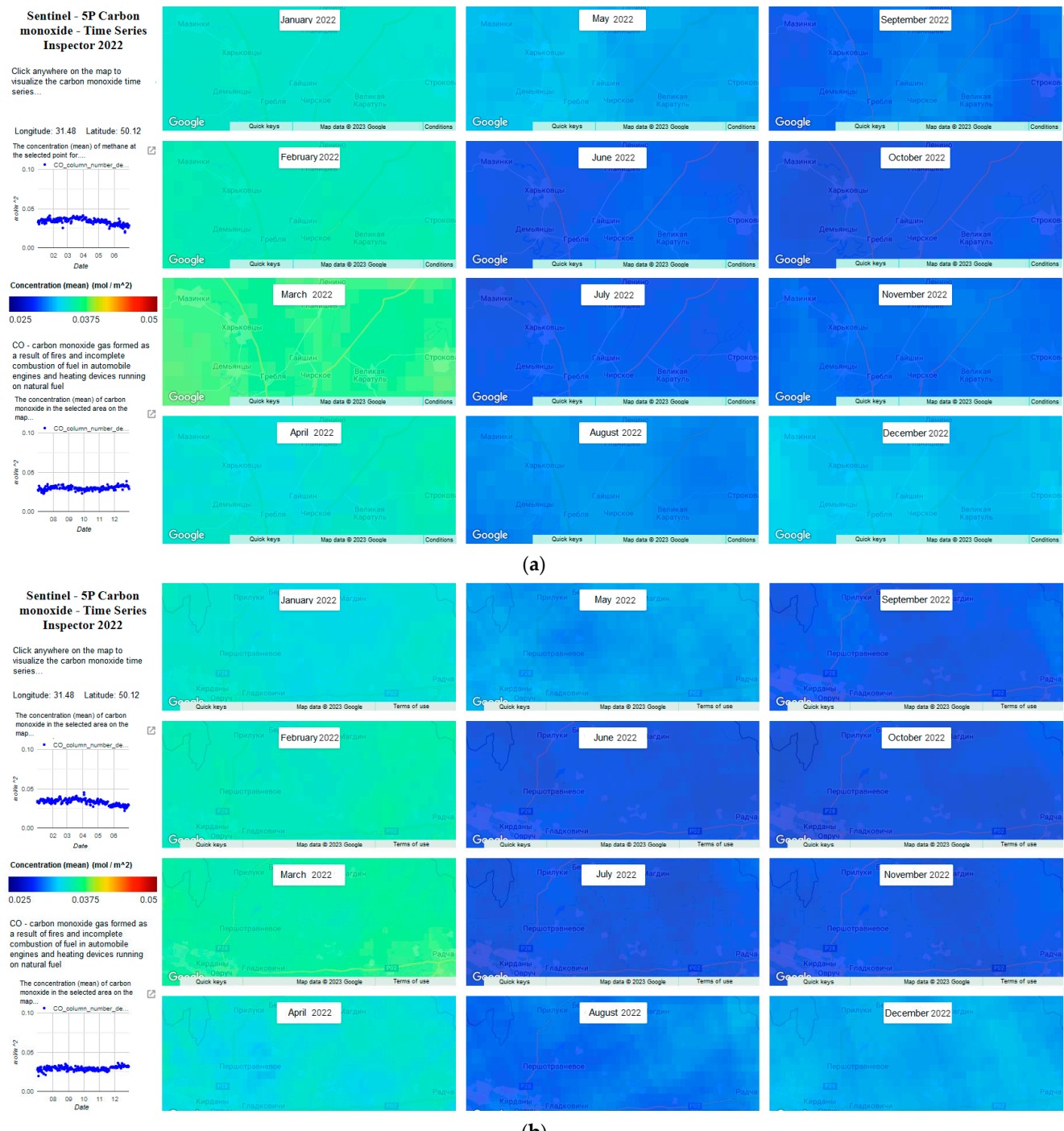

**Figure 11.** *Cont.*

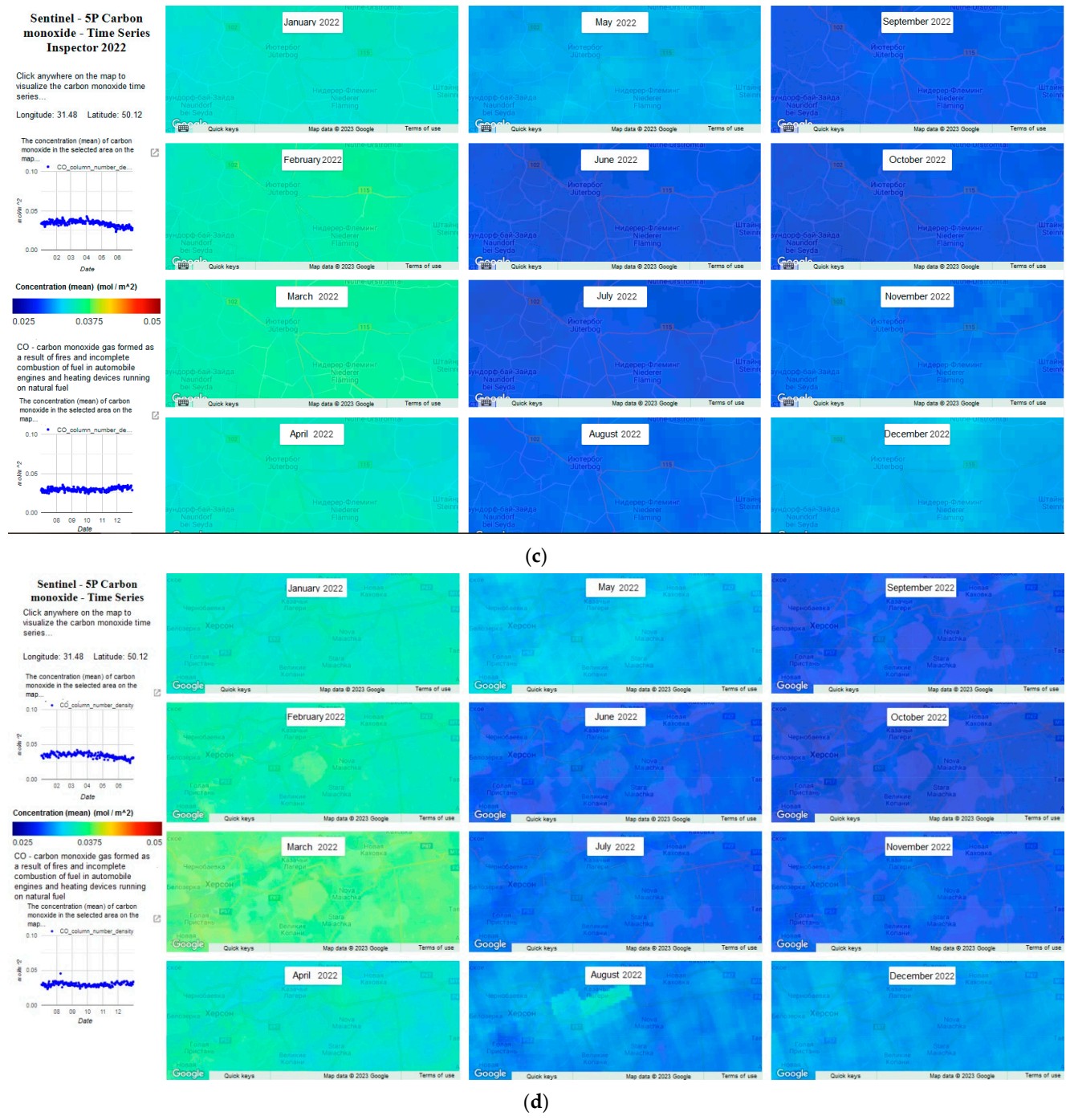

**Figure 11.** Results of CO emissions analysis during 2022 for the locations Gaishin (**a**), Ovruch (**b**), Jüterbog (**c**), and Oleshky Sands (**d**).

We developed our scientific and methodological apparatus taking into account the following approaches. The determination of promising raw materials for biofuels was carried out on the basis of an assessment of the emissions of two gases: methane $CH_4$ and carbon monoxide CO.

Assessment of methane emissions. During the study of the Oleshky Sands location, significant methane emissions were recorded that specifically arose there, which were not due to the decomposition of plants during the year; a comparison was made with an additional Novokamenka site, where the soils are similar but irrigated agriculture is carried out and, accordingly, parts of the vegetation rot directly in the fields. The results are shown in Figure 12.

**Table 3.** Recorded carbon monoxide emissions for the Gaishin.and Ovruch locations by year, mol/m$^2$.

| | Gaishin | | | | Ovruch | | | |
|---|---|---|---|---|---|---|---|---|
| | **2020** | **2021** | **2022** | **2023** | **2020** | **2021** | **2022** | **2023** |
| 1 | 0.0355 | 0.0386 | 0.0347 | 0.0324 | 0.0359 | 0.0381 | 0.0343 | 0.0320 |
| 2 | 0.0359 | 0.0382 | 0.0354 | 0.0352 | 0.0365 | 0.0377 | 0.0351 | 0.0345 |
| 3 | 0.0377 | 0.0395 | 0.0374 | 0.0342 | 0.0374 | 0.0384 | 0.0363 | 0.0328 |
| 4 | 0.0424 | 0.0382 | 0.0347 | 0.0338 | 0.0418 | 0.0371 | 0.0341 | 0.0337 |
| 5 | 0.0343 | 0.0330 | 0.0321 | 0.0329 | 0.0340 | 0.0324 | 0.0311 | 0.0324 |
| 6 | 0.0307 | 0.0307 | 0.0289 | 0.0354 | 0.0305 | 0.0301 | 0.0283 | 0.0333 |
| 7 | 0.0285 | 0.0330 | 0.0288 | | 0.0276 | 0.0325 | 0.0280 | |
| 8 | 0.0298 | 0.0402 | 0.0308 | | 0.0293 | 0.0380 | 0.0301 | |
| 9 | 0.0344 | 0.0380 | 0.0289 | | 0.0337 | 0.0375 | 0.0287 | |
| 10 | 0.0346 | 0.0348 | 0.0285 | | 0.0341 | 0.0348 | 0.0280 | |
| 11 | 0.0346 | 0.0333 | 0.0290 | | 0.0334 | 0.0331 | 0.0285 | |
| 12 | 0.0362 | 0.0345 | 0.0321 | | 0.0356 | 0.0343 | 0.0320 | |

**Table 4.** Recorded CO emissions for the Jüterbog and Oleshky Sands locations by year, mol/m$^2$.

| | Jüterbog | | | | Oleshky Sands | | | |
|---|---|---|---|---|---|---|---|---|
| | **2020** | **2021** | **2022** | **2023** | **2020** | **2021** | **2022** | **2023** |
| 1 | 0.0355 | 0.0384 | 0.0347 | 0.0332 | 0.0362 | 0.0389 | 0.0351 | 0.0322 |
| 2 | 0.0370 | 0.0382 | 0.0357 | 0.0341 | 0.0368 | 0.0377 | 0.0359 | 0.0356 |
| 3 | 0.0380 | 0.0381 | 0.0365 | 0.0336 | 0.0382 | 0.0399 | 0.0372 | 0.0350 |
| 4 | 0.0384 | 0.0379 | 0.0356 | 0.0343 | 0.0399 | 0.0386 | 0.0348 | 0.0344 |
| 5 | 0.0347 | 0.0330 | 0.0314 | 0.0340 | 0.0348 | 0.0334 | 0.0321 | 0.0332 |
| 6 | 0.0309 | 0.0302 | 0.0287 | 0.0341 | 0.0311 | 0.0311 | 0.0294 | 0.0353 |
| 7 | 0.0281 | 0.0328 | 0.0288 | | 0.0296 | 0.0337 | 0.0298 | |
| 8 | 0.0301 | 0.0403 | 0.0297 | | 0.0305 | 0.0402 | 0.0312 | |
| 9 | 0.0343 | 0.0371 | 0.0290 | | 0.0337 | 0.0388 | 0.0288 | |
| 10 | 0.0342 | 0.0342 | 0.0284 | | 0.0331 | 0.0357 | 0.0283 | |
| 11 | 0.0338 | 0.0331 | 0.0299 | | 0.0341 | 0.0330 | 0.0296 | |
| 12 | 0.0363 | 0.0336 | 0.0321 | | 0.0358 | 0.0340 | 0.0311 | |

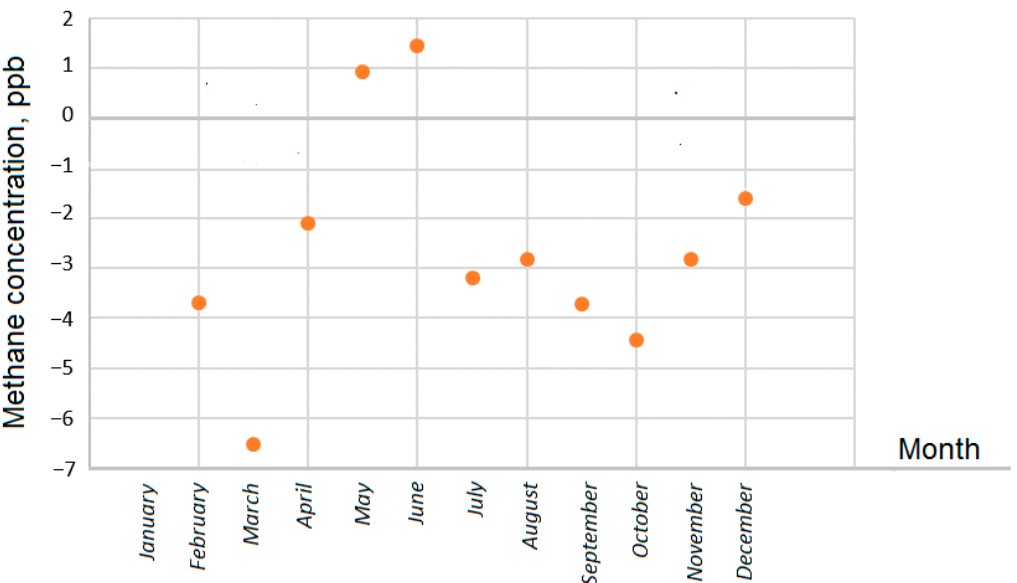

**Figure 12.** Dependence of the averaged difference in methane emissions for 2020–2023 between the Oleshky Sands and Novokamenka sites.

Based on the data provided, emissions at the site in the desert were consistently higher than those at the sites with agricultural land only for 2 months—May and June—that is,

when there was increased green biomass in the fields. This phenomenon can be explained by a combination of different mechanisms of methane formation, which is due to the decomposition of vegetation, and emissions from the depths of the earth's surface.

The assumed organic origin of the methane coincides with the data presented in the works of W. Takeuchi et al. [28] on methane emissions in the tundra and M. Jia et al. [29] on methane emissions in rice fields. An alternative origin of methane from the subsoil, i.e., anthropogenic sources, was also recorded in studies on the needs of the oil and gas industry by J.-F. Gauthier [30]. A somewhat different interpretation of the results of satellite monitoring is shown in the work of F. Wang et al. [31] where, for the Middle East region with significant hydrocarbon reserves, a significant impact from emissions of plant origin was also noted.

A possible explanation for the results in Figure 10 is the complex mechanism underlying methane interaction with plants, which can prevent its emission into the atmosphere through various mechanisms and, accordingly, this issue requires additional research. Figure 13 shows the averaged data for the Ovruch, Jüterbog, and Gaishin sites.

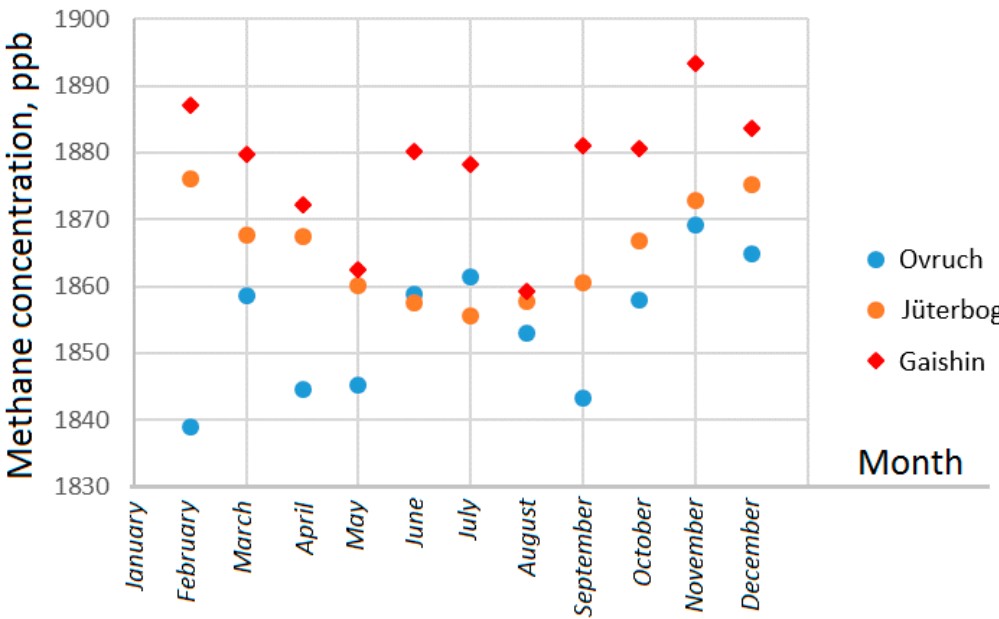

**Figure 13.** Dependence of the averaged 2020–2023 differences in CH$_4$ emissions between the Ovruch, Jüterbog, and Gaishin sites.

For each site, the dependence on methane emissions changes during the year has a different nature. If for Gaishin emissions are approximately the same throughout the year, then for Ovruch they gradually increase throughout the year, and for Jüterbog the dependence is parabolic with a minimum level in the middle of the year. The explanation for the gradual increase in emissions for Ovruch can be justified by the decay of biomass, which gradually increases and, as a result of the implementation of crop-growing practices, the amount reaches a maximum level in October. Similar results were obtained for Novokamenka. For these objects, linear and exponential equations were used in the approximation; the coefficient of approximation reliability was approximately the same, and was 0.56 for Ovruch and 0.38 for Novokamenka, respectively.

After a decrease in temperature, the processes of decay in plant residues slow down and, accordingly, the minimum indicators occur in winter. For Jüterbog, methane emissions are, on the contrary, at a maximum in winter and at a minimum in summer; such a nature of dependence may correspond not to the crop-growing, but to the livestock-growing orientation of the local agricultural sector. According to the data provided by B. He

et al. [32], emissions from livestock farms in China are so significant that the emission plume is recorded even by satellites. Accordingly, in winter livestock is indoors, and in summer outdoors, which is reflected in the methane emissions.

For Gaishin, methane emissions are greater than in the Ovruch and Jüterbog areas, which is probably a consequence of the simultaneous influence of many factors: methane emissions from crop-growing practices, livestock farming, etc.

Compared to other greenhouse gases such as carbon monoxide, methane is relatively poorly diffusible and, accordingly, its source can be traced along its distribution plume, as shown by open access data from the Sentinel-2 satellite in the work of T. Ehret et al. [33]. The pattern recognition technique for methane emissions was improved in the work of E. Ouerghi et al. [34], where data from the Prisma satellite were used, taking into account meteorological data on wind strength and direction in relation to the distribution plume of the reagent.

Assessment of carbon monoxide emissions. Unlike methane, CO has significantly fewer possible formation mechanisms; however, it mixes better with atmospheric air, which complicates the identification of its sources. Figure 14 shows the results of satellite measurements of CO content over the territory of Ukraine and Poland.

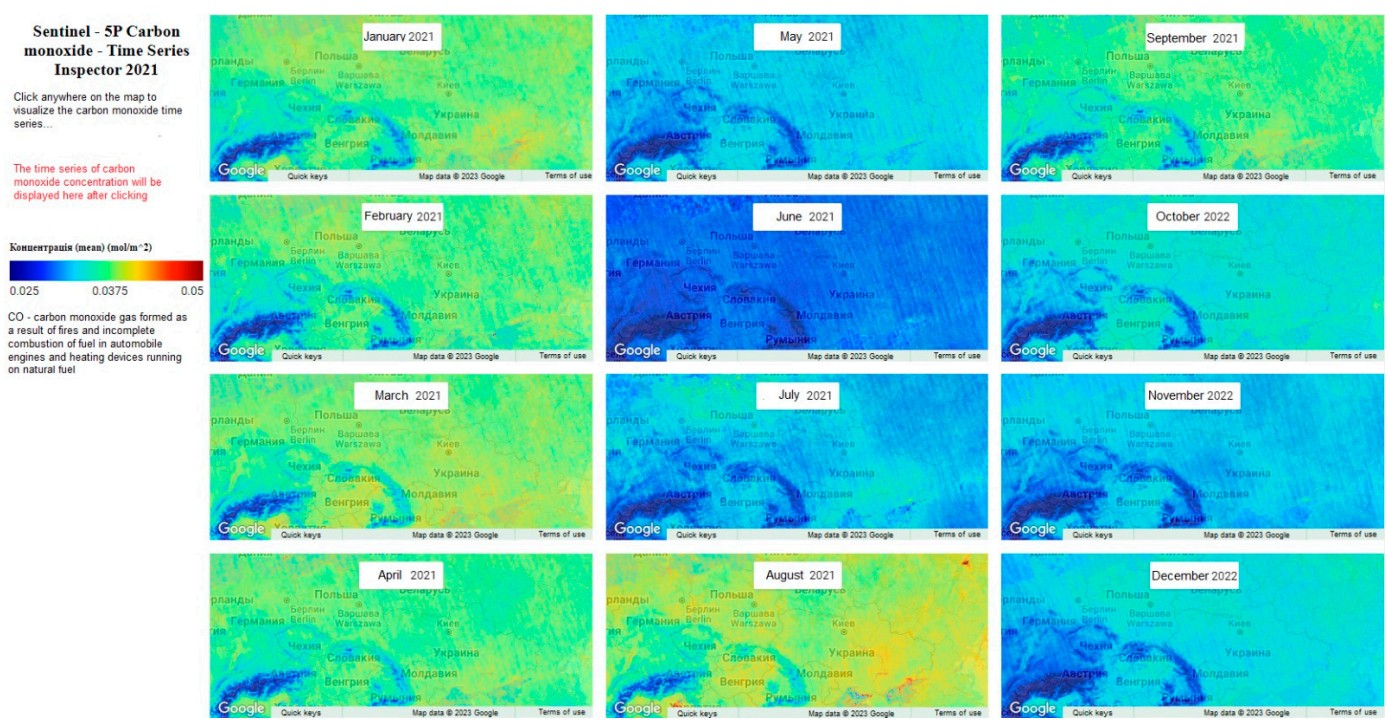

**Figure 14.** Distribution of CO emissions over the territory of Ukraine and Poland in 2021.

In Figure 13, the zones with low CO content correspond to the Carpathian and Alpine mountains, while the distribution of this greenhouse gas is relatively uniform in the rest of the territory. The fixation of high emissions in late winter and early spring coincides with the data obtained in the work of H. Yuan et al. [35] in urbanized areas of China. A promising future direction may be to compare CO emissions in different areas (Figure 15).

As can be seen from the data presented in Tables 3 and 4, the nature of the changes in CO emissions is fundamentally the same for all experimental plots, and the largest difference is recorded in the spring–summer period. The results obtained are quite expected. This can be attributed to indirect signs of the influence of agricultural practices on these processes. Figure 16 shows the results of the difference in indicators when Gaishin was allocated as the base option and the indicators of all others were subtracted from it.

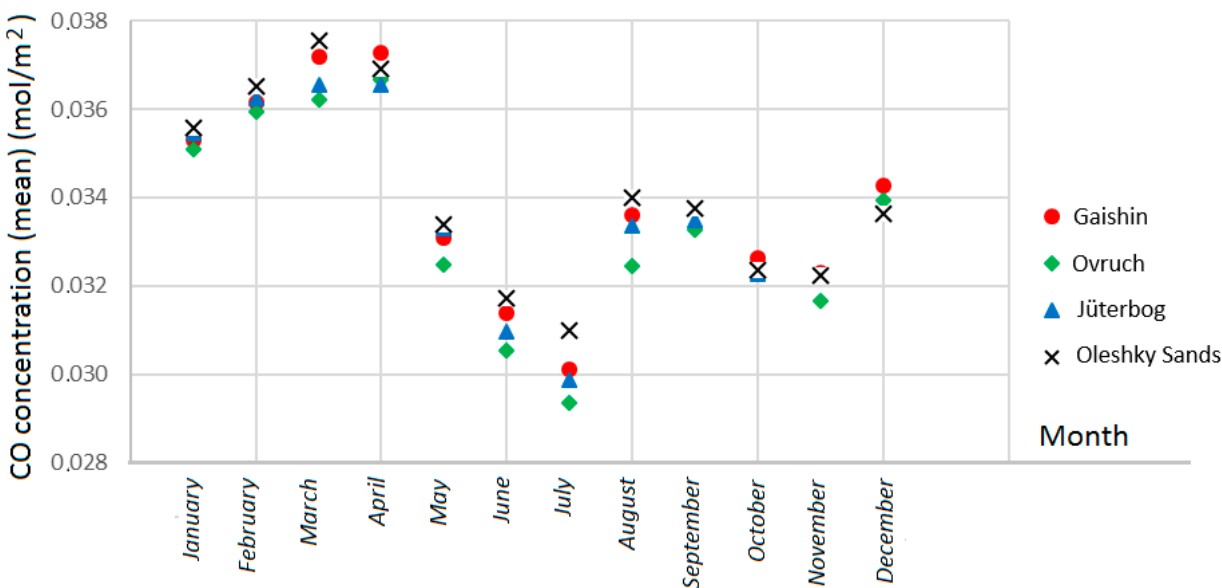

**Figure 15.** Averaged over the observation period, changes in CO emissions for the research sites during the year.

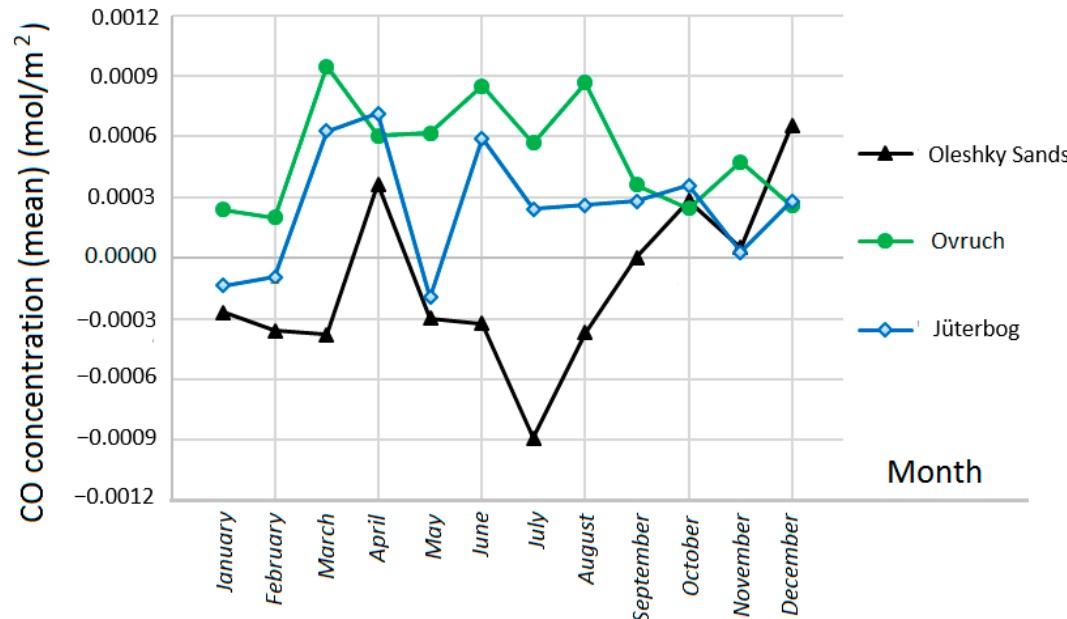

**Figure 16.** Difference in average CO emissions between the base location Gaishin and Oleshky Sands, Ovruch, and Jüterbog, respectively.

The data obtained show that at the Oleshky Sands site, CO emissions are higher throughout almost the entire year compared to the other sites. This situation, together with the methane emissions, corresponds to the results obtained in the work of R. P. Singh and S. Sarkar [36] in the Aliso Canyon site, where, along with an emergency methane release, CO emissions also occurred. Accordingly, such combined emissions may be anthropogenic in nature and, accordingly, such sites are not advisable to use when comparing locations for the assessment of promising organic raw materials for biogas production [37].

According to the data presented in Figure 17 regarding CO emissions, emissions in Gaishin are the highest, but the results are primarily indicative. The observed CO emissions may also be due to the different quality of fuel used for internal combustion engines, differences in the design and quality of engine performance, etc. For an approximate estimate, we proposed to use the well-known case of a fire in the Chernobyl zone in 2021,

when approximately 11.5 thousand hectares of forest burned in 10 d. Taking as a base value a volume of 235 $m^3$ of wood per hectare, the amount of burned forest was estimated at 2.7 million $m^3$ of wood.

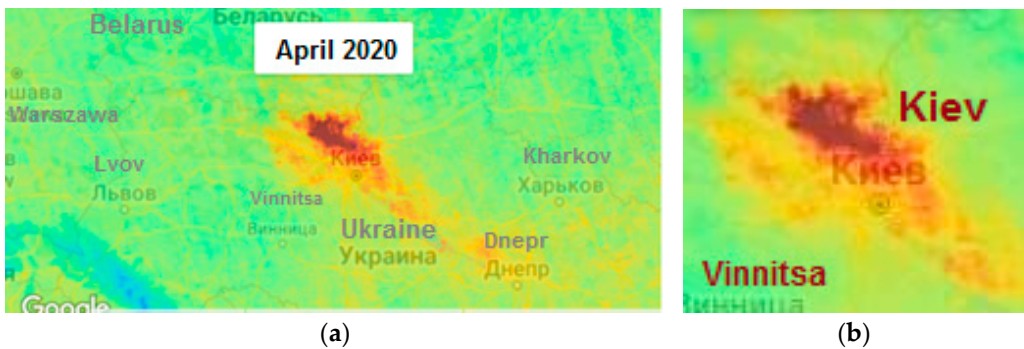

**Figure 17.** CO emissions in April 2020 due to a fire in the Chernobyl zone (**a**) and a test site to assess the impact of forest burning on an increase in CO content on the (**b**).

For the analysis, an area of 12 million ha was selected and, using the method of decoding the data encoded in the form of a palette, the average value of CO emissions over the territory was calculated, which totaled 0.043 $mol/m^2$. In parallel, the average values at the location for April during 2021–2023 were calculated, which were determined as 0.038, 0.037, and 0.036 $mol/m^2$, respectively. Thus, an increase in the CO concentration of 0.006 $mol/m^2$ over an area of 12 million ha was the result of the combustion of 2.7 million $m^3$ of wood. According to the data shown in Figure 14, in Gaishin during the summer the CO concentration is higher than in Ovruch and Jüterbog by 0.009 and 0.004 $mol/m^2$, respectively. Such values give reason to assume that the Gaishin location has large volumes of biomass that are promising for biogas production. This article presents an improvement of the methods and algorithms that were previously applied to another task—identifying stable man-made areas with thermal emissions from satellite images [38]. They can be used for the sustainable development of territories and crop production, including the cultivation of energy crops.

The materials in this study are intended to be used in the field for the robotic processing and collection of agricultural raw materials [39], the robotic cleaning of biogas tanks (methane tanks) [40], in the design of control systems for biogas reactors [41–43], and for forecasting the yield of raw materials for biogas production using intelligent methods [44–46]. The results of this study can also be used in various scientific and practical areas in accordance with the European Green Deal [47,48]. In particular, for the development of hydrogen energy [49], carbon footprint analysis in agriculture [50], algae biomass analysis [51], the assessment of anthropogenic emissions from brown coal flue gases [52], and the valorization of Spirodela polyrrhiza biomass [53] and anaerobic digestate [54] for the production of biofuels for distributed energy.

## 4. Conclusions

1. Identification of unused biomass is proposed using objective data from the assessment of $CH_4$ and CO in the atmosphere based on satellite monitoring results.

2. Based on experimental studies conducted using space satellites around the Earth, it has been established that the location of the territory Gaishin has better prospects for collecting plant raw materials for biogas production than the location of the territorial area of the city of Ovruch, in which emissions are significantly lower. From March 2020 to August 2023, a CO concentration of 0.0009 $mol/m^2$ higher was recorded in Gaishin than in Ovruch, which is explained precisely by crop growing practices.

3. To determine unused/lost biomass, it is advisable to consider the emission concentrations in different periods, for example, CO emissions in spring may be due to the burning of plant residues in the fields, and a sharp increase in $CH_4$ emissions in October–November may be due to the decay of plant biomass. That is, there are prospects for identifying the sources of unused biomass.

**Author Contributions:** Conceptualization, O.O. and S.S.; methodology, S.S., T.H., and F.H.; funding acquisition, T.N. and S.G.; resources, T.H. and S.G.; writing—original draft, O.O.; writing—review and editing, N.K.; software, R.P. and F.H.; validation, N.K.; data curation, N.K. and M.M.; formal analysis, M.S.; supervision, T.H. All authors have read and agreed to the published version of the manuscript.

**Funding:** The publication was (co)financed by the science development fund at the Warsaw University of Life Sciences—SGGW.

**Institutional Review Board Statement:** Not applicable.

**Informed Consent Statement:** Not applicable.

**Data Availability Statement:** The data presented in this study are available on request from the corresponding author.

**Acknowledgments:** The anonymous reviewers are gratefully acknowledged for their constructive reviews that significantly improved this manuscript, as well as the International Visegrad Fund (https://www.visegradfund.org, accessed on 13 January 2025) and Ukrainian University in Europe (https://universityuue.com/, accessed on 13 January 2025).

**Conflicts of Interest:** The authors declare no conflicts of interest.

## Appendix A

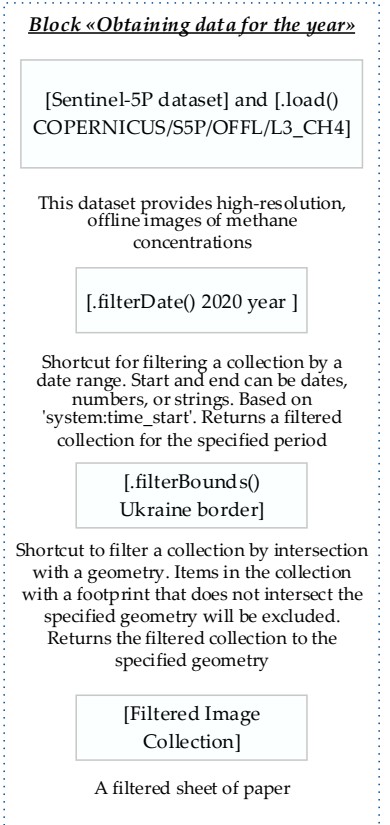 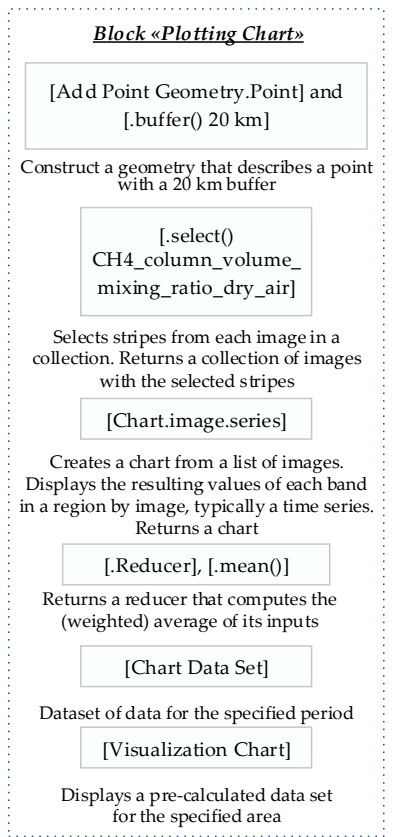 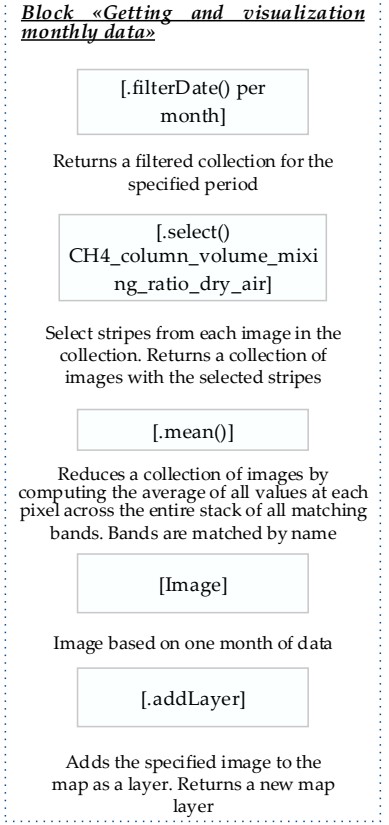

**Figure A1.** Example of Working with Coding.

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
