# Peer review of "Evaluation of Promising Areas for Biogas Production by Indirect Assessment of Raw Materials Using Satellite Monitoring"

_sustainability, doi:10.3390/su17052098_

Round 1
Reviewer 1 Report
Comments and Suggestions for Authors
1. Abstract: Lack of specific quantitative data
The abstract lacks specific quantitative data, which weakens the persuasiveness of the research conclusions. It is recommended to include numerical values for methane and carbon monoxide emissions in different regions and specify the key parameters measured in the experiments. This will help to present the actual impact of the research findings more clearly.
2. Introduction: Reorganization
The current structure of the manuscript can be improved by integrating the two separate Literature Review sections into the Introduction to provide a more coherent and logically structured background. The introduction should be reorganized following a clear why, how, who, and what framework.
3. Introduction: Roadmap
The introduction would benefit from a roadmap or schematic overview at the end of the introduction to provide a clear structure of the research workflow. This would help guide the reader through the study's logical progression and methodology.
4. Results and Discussion: Minimizing Square Brackets and Relocating Technical Details
Excessive use of square brackets disrupts readability. Move software details (e.g., dataset loading, function usage) to Supporting Information or an appendix. In the main text, integrate descriptions naturally. Example (lines 293-294): The Sentinel-5P dataset was used for high-resolution methane imagery, filtering data from 2020.
This improves clarity while keeping technical details accessible.
5. Conclusion: Improves with Concise Restatement
1) The conclusion should be more concise and focused on key findings. Avoid restating the methodology and emphasize the significance of results.
2) Summarize Findings Clearly—Instead of describing the method, highlight its impact on biogas production planning.
3) Provide Comparative Data—Specify how much higher emissions in Gaishyn were compared to Ovruch.
4) Clarify Practical Implications—Explain how integrating CHâ‚„ and CO improves biomass assessment.
6. Figure Formatting: Need better resolution, axis labeling, and formatting for readability.
1) Increase Resolution & Remove Grid—Ensure high-quality images and remove background grids for clarity.
2) Fix Axis Labels—Move X-axis labels to the bottom and replace numbers (1-12) with month names. Use correct Y-axis unit formatting (e.g., "mol/m²" instead of "mol/m^2").
3) Standardize Legend—Remove unnecessary symbols (e.g., "Γ") and ensure clear, consistent labeling.
7. Verify and Reduce Word Count (≤7500 Words)
Ensure the manuscript adheres to the 7500-word limit, including figures and tables.
Author Response
Dear Reviewer, Thank you for your valuable comments. We have tried to take them into account and are providing a new version of the article.
Question 1. Abstract: Lack of specific quantitative data
The abstract lacks specific quantitative data, which weakens the persuasiveness of the research conclusions. It is recommended to include numerical values for methane and carbon monoxide emissions in different regions and specify the key parameters measured in the experiments. This will help to present the actual impact of the research findings more clearly.
Answer 1. The abstract was supplemented with the words:
"During the period from March to August 2020-2023, a higher CO concentration was recorded on average by 0.0009 mol/m2, which is explained precisely by crop growing practices. Thus, in the desert (Oleshkivskie Pisky), large methane emissions were recorded throughout the year, which could not be explained by crop growing practices or the livestock industry."
Question 2. Introduction: Reorganization
The current structure of the manuscript could be improved by combining the two separate Literature Review sections into the Introduction to provide a more coherent and logically structured framework. The Introduction should be reorganized to follow a clear structure of “why,” “how,” “who,” and “what.”
Answer 2. Done, combined sections, removed generalities, and kept specifics.
Question 3. Introduction: Roadmap
The Introduction would benefit from a roadmap or flow chart at the end of the introduction to provide a clear structure of the research workflow. This would help the reader navigate through the logical flow and methodology of the study.
Answer 3. Done, added new Figure 1 with a flow chart of the study.
Question 4. Results and Discussion: Minimize square brackets and move technical details
Excessive use of square brackets hinders readability. Move software details (e.g. dataset download, function usage) to the Supporting Information section or Appendix. Integrate descriptions naturally into the main text. Example (lines 293–294): The Sentinel-5P dataset was used to obtain high-resolution methane images, filtering out data from 2020.
This improves clarity while keeping the technical details accessible.
Answer 4: Fixed. Moved software code to the Appendix section, presented as a diagram.
Question 5: Conclusion: Improved with a short summary
The conclusion should be more concise and focused on the key findings. Avoid repeating the methodology and emphasize the significance of the results.
Summarize the findings clearly — instead of describing the method, emphasize its impact on biogas production planning.
Provide comparative data — indicate how much higher the emissions were in Gaishin compared to Ovruch. .
Clarification of practical implications — Explain how integrating CHâ‚„ and CO improves biomass assessment.
Answer 5. Corrected, paragraphs shortened and specified, numerical results added
Identification of unused biomass is proposed based on objective data on the assessment of methane and CO in the atmosphere based on the results of satellite monitoring
Conclusions corrected.
Added text:
During March to August 2020-2023, a higher concentration of CO in Geyshina was recorded in Ovruch by an average of 0.0009 mol/m2, which is explained precisely by crop production practices.
Added paragraph:
To determine unused/lost biomass, it is advisable to consider emission concentrations in different periods, for example, CO emissions in the spring can be caused by the burning of plant residues in the fields, and a sharp increase in methane emissions in October-November - by rotting plant biomass. There are prospects for identifying sources of unused biomass identification.
Question 6. Figure formatting: Need better resolution, axis labels, and formatting for readability.
1) Increase resolution and remove gridlines - Ensure high quality images and remove background gridlines for clarity.
Answer: Increased resolution, and original figures will be sent to the editors for posting at full size.
Regarding the coordinate grid, we think it is useful in these graphs as it improves the definition of the coordinates of the monthly concentration value point.
2) Correct axis labels - Move X-axis labels down and replace numbers (1-12) with month names. Use correct formatting of Y-axis units (e.g. "mol/m²" instead of "mol/m^2").
Answer : Corrected, labeled months, moved X-axis down, corrected Y-axis label.
3) Standardize legend - remove unnecessary symbols (e.g. "Γ") and provide clear, consistent labeling.
Answer: Corrected.
Question 7. Review and reduce word count (≤7500 words)
Ensure manuscript complies with 7500 word limit, including figures and tables.
Answer 7: Done, reduced text from 8972 words to 7534 words.
Reviewer 2 Report
Comments and Suggestions for Authors
A pressing issue in the sustainable development of agricultural engineering today is the utilization of biogas equipment to generate electricity and heat from organic waste produced in agricultural activities and low-quality products, which also contributes to improving environmental safety. The authors monitored the biogenic methane emissions from decomposing organic waste into the atmosphere and conducted a comparative assessment of carbon monoxide emissions from burning agricultural waste across different regions of the area. I find this particularly intriguing, as the conclusions drawn by the authors through their analysis are of significant importance to related research. The manuscript demonstrates strong logical coherence and thorough argumentation, with only minor issues that could benefit from appropriate revisions.
-
The author has made errors in the notation of certain chemical symbols in the manuscript, such as the CO2 in lines 475 and 484, which should be corrected to CO2. It is important for the author to thoroughly check and revise the entire manuscript.
-
Some units in the manuscript should be abbreviated; for example, "days" in line 509 should be changed to "d." It is recommended that the entire manuscript be reviewed and corrected accordingly.
-
There are issues with the notation of area units in the manuscript, particularly in lines 509, 510, 517, 518, and 520, where the use of m2 and m3 is incorrect.
-
The citation format for some references in the manuscript is inappropriate, specifically in lines 418, 419, 421, and 422. These should be presented in a narrative style with citations placed at the end of the sentences. A thorough review and revision of the entire manuscript is advised.
-
It is recommended that the clarity of Figure 5 be improved to meet the journal's requirements.
Author Response
Dear Reviewer, Thank you for your valuable comments. We have tried to take them into account and are providing a new version of the article.
Question 1. The author has made errors in the notation of some chemical symbols in the manuscript, such as CO2 in lines 475 and 484, which should be corrected to CO 2. It is important for the author to carefully check and revise the entire manuscript.
Answer 1. Corrected throughout the text CO2 to CO.
Question 2. Some units of measurement in the manuscript should be abbreviated; for example, "days" in line 509 should be changed to "d". It is recommended that the entire manuscript be reviewed and corrected accordingly.
Answer 2. Corrected
Question 3. There are problems with the notation of units of area in the manuscript, particularly in lines 509, 510, 517, 518, and 520, where the use of m2 and m3 is incorrect.
Answer 3. Corrected throughout the text.
Question 4. The citation format of some references in the manuscript is inappropriate, particularly lines 418, 419, 421, and 422. They should be presented in a narrative style with citations placed at the end of sentences. A thorough review and revision of the entire manuscript is recommended.
Answer 4. Corrected, citations brought to a uniform format
Question 5. It is recommended to improve the clarity of Figure 5 so that it meets the requirements of the journal.
Answer 5. Corrected, the quality of the figure has been improved, the resolution is higher. The original figure will be sent to the editors for posting in high resolution.
Reviewer 3 Report
Comments and Suggestions for Authors
Data for biomethane and monoxide emissions is important for monitoring rotting organic waste and burning agricultural waste. It is a worthy study. However, there are some flaws in the paper.
The title can not reflect your research. There is no relation with biogas plants in this study.
Some of the language is a bit verbose. For example line 35 “emissions were assessed by comparing them for different areas of the area.”
Emphasize originality in the Introduction part.
The analytical method should be more detailed.
Fig. 5 the picture is blurry, please make it clear.
Table 1, Recorded CH4 methane emissions. You can delete CH4 or methane.
Data from Fig 10 are repeat with the data in some Tables. Could you arrange the data properly? So did in Fig. 11.
Author Response
Dear Reviewer, Thank you for your valuable comments. We have tried to take them into account and are providing a new version of the article.
Question 1. The title cannot reflect your study. There is no connection with biogas plants in this study.
Answer 1. We have proposed a new title: Evaluation of promising areas for biogas production by indirect assessment of raw materials using satellite monitoring
Question 2. Some of the wording is a bit wordy. For example, line 35 "emissions were estimated by comparing them for different areas of the area."
Answer 2. The phrase has been removed.
Question 3. Emphasize the originality of the introduction.
Answer 3. The following has been added to the abstract:
Traditional methods of assessing the available raw materials of plant origin based on statistics from farms have proven ineffective. A hypothesis has been put forward about the possibility of assessing the available biomass by indirect signs due to their destruction, namely the assessment of methane and CO emissions according to satellite monitoring data.. Highlighted.
Question 4. The analytical method should be more detailed.
Answer 4. The method is described in more detail in the article [23] with the participation of our authors, it is referenced 3 times. Also, the program code with explanations is presented as a diagram in the "Appendix" section, which is more visual. Also, Fig. 1 with the research scheme has been added to the introduction.
S.A. Shvorov, N.A. Pasichnyk, O.A. Opryshko, D.S. Komarchuk, A.O. Dudnyk and F.V. Hluhan, "The Methodological Foundations of Building an Energy Efficient Community," 2022 IEEE 16th International Conference on Advanced Trends in Radioelectronics, Telecommunications and Computer Engineering (TCSET), Lviv-Slavske, Ukraine, 2022, pp. 297-300, doi: 10.1109/TCSET55632.2022.9766956.
Question 5. Fig. 5. The image is blurry, make it clearer.
Answer 5. Corrected, the quality of the figure has been improved, the resolution is higher. The original figure will be sent to the editors for posting in high resolution.
Question 6. Table 1, Registered emissions of methane CH4. You can remove CH4 or methane.
Answer 6. Corrected, "methane" left
Question 7. The data from Fig. 10 is repeated with the data in some tables. Could you please arrange the data correctly? This is what we have done in Fig. 11.
Answer 7. These figures with graphs show some of the information from the tables more clearly, the tables contain more complete data. However, if the tables look cumbersome, we can put them in the appendix.
Reviewer 4 Report
Comments and Suggestions for Authors
Before the manuscript can be published, in my opinion it requires some minor revisions.
- Line 121. Before an abbreviation appears in the body of the manuscript, it should be explained. Check all abbreviations in the manuscript.
- There are two figures numbered 5 in the manuscript. Correct the numbering of the figure titles and the numbering in the body of the text from page 10.
- Tables 3 and Table 4. Provide a more detailed discussion of the results. Are similar values of carbon monoxide IV and carbon monoxide concentrations a surprise or expected results?
- Line 428. “In Fig. Figure 11…”
- General note: Use periods instead of commas to indicate decimal places in numbers (both in tables and figures).
- In Figure 7, make sure that all negative values have a minus sign next to them, not just the seven value.
- You have used a mixed system of citing references in the manuscript, at the beginning the name, year and number with brackets, e.g. Xintan Zhang et al (2023) in [20], at the end of the work only square brackets. Standardize your references citation system.
Author Response
Dear Reviewer, Thank you for your valuable comments. We have tried to take them into account and are providing a new version of the article.
Question 1. Line 121. Before an abbreviation appears in the text of the manuscript, it should be explained. Check all abbreviations in the manuscript.
Answer 1. Corrected to "in each locality or district", and other abbreviations explained.
Question 2. There are two figures numbered 5 in the manuscript. Correct the numbering of the figure titles and the numbering in the text from page 10.
Answer 2. Corrected, numbering and references brought into line
Question 3. Tables 3 and 4. Provide a more detailed discussion of the results. Are the similar values ​​of carbon monoxide and carbon monoxide concentration an unexpected or expected result?
Answer 3. Answer added. Highlighted.
(As can be seen from the data presented in Tables 3 and 4, the nature of changes in CO emissions is fundamentally the same for all experimental plots and the largest difference is recorded in the spring-summer period. The results obtained are entirely expected)
Question 4. Line 428. "In Fig. Fig. 11..."
Answer 5. Corrected
Question 5. General note: Use periods instead of commas to indicate decimal places in numbers (both in tables and figures).
Answer 5. Corrected in figures and in tables
Question 6. In Figure 7, make sure that all negative values ​​have a minus sign, and not just the value "seven".
Answer 6. Corrected
Question 7. You used a mixed system of citing references in the manuscript, at the beginning the name, year and number in brackets, for example, Xintan Zhang et al (2023) in [20], at the end of the work only square brackets. Standardize your system of citing references.
Answer 7. Corrected, citation brought to a uniform format